# Topology-Aware Uncertainty for Image Segmentation

**Saumya Gupta**[1]*   **Yikai Zhang**[2]   **Xiaoling Hu**[1,3]   **Prateek Prasanna**[1]   **Chao Chen**[1]

[1]Stony Brook University, NY, USA    [2]Morgan Stanley, NY, USA
[3]Athinoula A. Martinos Center for Biomedical Imaging,
Massachusetts General Hospital and Harvard Medical School, MA, USA

## Abstract

Segmentation of curvilinear structures such as vasculature and road networks is challenging due to relatively weak signals and complex geometry/topology. To facilitate and accelerate large scale annotation, one has to adopt semi-automatic approaches such as proofreading by experts. In this work, we focus on uncertainty estimation for such tasks, so that highly uncertain, and thus error-prone structures can be identified for human annotators to verify. Unlike most existing works, which provide pixel-wise uncertainty maps, we stipulate it is crucial to estimate uncertainty in the units of topological structures, e.g., small pieces of connections and branches. To achieve this, we leverage tools from topological data analysis, specifically discrete Morse theory (DMT), to first capture the structures, and then reason about their uncertainties. To model the uncertainty, we (1) propose a joint prediction model that estimates the uncertainty of a structure while taking the neighboring structures into consideration (inter-structural uncertainty); (2) propose a novel Probabilistic DMT to model the inherent uncertainty within each structure (intra-structural uncertainty) by sampling its representations via a perturb-and-walk scheme. On various 2D and 3D datasets, our method produces better structure-wise uncertainty maps compared to existing works. Code available at https://github.com/Saumya-Gupta-26/struct-uncertainty

## 1   Introduction

Curvilinear segmentation is an essential initial step in various medical and non-medical applications, involving the precise extraction of fine-scale structures, such as blood vessels, nerves, and other elongated objects [21, 33]. For example, extraction of retinal vasculature is an essential precursor to understanding disease progression and assessing therapeutic effects [15]. In civil engineering, road network and railway track segmentation can support urban planning and transportation system optimization [45]. Despite the success of deep learning [5, 6, 25, 37], automatic segmentation of thin structures remains challenging due to their relatively low visibility and complex topology. Existing segmentation methods often make topological errors such as broken connections or missing branches.

As a cost-efficient alternative, in many applications, one employs semi-automatic techniques, e.g., iterative proofreading by human annotators [23]. On the other hand, proofreading of complex fine-scale structures can be extremely time-consuming [53]. This necessitates a better strategy to direct the annotators' attention towards locations that are more error-prone. Following the classic active learning principle [18, 55, 61], we may estimate the *uncertainty* [19] and concentrate on the locations where a neural network is the least certain.

Despite many existing studies on segmentation uncertainty [12, 51, 60], most existing uncertainty estimation methods do not apply to curvilinear structure segmentation. Existing methods typically generate pixel-wise uncertainty maps. Such maps highlight pixels along the boundary of all structures,

---

*Email: saumgupta@cs.stonybrook.edu

37th Conference on Neural Information Processing Systems (NeurIPS 2023).

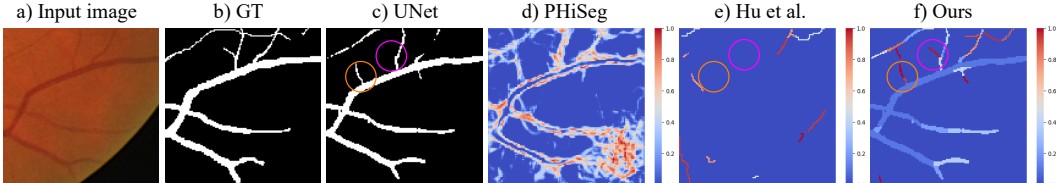

| a) Input image | b) GT | c) UNet | d) PHiSeg | e) Hu et al. | f) Ours |

Figure 1: Motivating examples for structure-wise uncertainty. In the segmentation result (c), orange highlights a false positive structure, and pink highlights a false negative. Methods (d)-(f) are uncertainty estimates of the prediction in (c). PHiSeg [4] assigns pixels along boundaries as uncertain. Hu et al. [28] captures uncertainty at a structural level, but produces overconfident maps (assigns zero uncertainty to many structures). Ours produces better structure-wise uncertainty estimates: both the highlighted false positive/negative structures have high uncertainty.

regardless of their width or thickness (see Fig. 1(d)). This offers limited information for human annotators; a desirable uncertainty map should instead highlight the error-prone "structures", e.g., small vessels/branches or short stretches of roads that tend to be disconnected or missed.

In this paper, we propose a new topology-aware uncertainty estimation method that highlights error-prone structures as a whole (such as in Fig. 1(f)). By highlighting structures with high uncertainty, our method empowers annotators to accept or reject/correct structural proposals efficiently, thus streamlining the proofreading process. To capture the uncertainty of a given segmentation network's prediction at a structural level, we require the realization of two key components: a) decompose the prediction into a set of constituent structures, including false positives and false negatives, and b) estimate uncertainties of all the structures. Furthermore, we need to consider two types of structural uncertainty, intra-structural and inter-structural. The *intra-structural uncertainty* of a structure is due to its intrinsic composition, e.g., geometry, intensity, and the segmentation network's confidence. The *inter-structural uncertainty* is more contextual; it is due to interactions between neighboring structures. Our method explicitly models the two types of uncertainty.

Given a segmentation model, our method uses its likelihood map (Fig. 2(i)) plus the input image to estimate structural uncertainties. First, we obtain a structural decomposition of the prediction, i.e., a collection of one-pixel-wide pieces/structures (see Fig. 2(ii)). Each structure represents a potential branch/connection according to the segmentation model. We employ the principles of the classic discrete Morse theory (DMT) [14, 43]. Intuitively, DMT treats the likelihood function as a terrain function and extracts landscape features, e.g., mountain ridges or valleys, as structures. Note we are capturing all possible structures visible in the likelihood map, including the ones that do not appear in the segmentation due to low probability.

Next, we estimate uncertainties for all these structures. Existing uncertainty estimation approaches often sample multiple hypotheses and calculate the variance across them [17, 34, 69]. However, this principle is not feasible in our problem; with $N$ structures, the space of all their combinations is of exponential size ($2^N$). Sampling from such a space is very challenging. An alternative is to make independent uncertainty estimation on each structure. However, this is also suboptimal as it ignores inter-structural uncertainty. Fig. 2(vi) shows three false positive structures. Treating all structures independently will incorrectly assign the horizontal structure with low uncertainty (see Fig. 2(vii)).

We propose a joint inference model (i.e., a graph neural network [59]) to jointly predict uncertainties on all the structures. This joint inference framework avoids explicit enumeration/sampling over the exponential size space of hypotheses. It also takes into account the inter-structural uncertainty. In Fig. 2(viii), our method correctly assigns high uncertainties to all three false positive structures. To supervise the training process, we use the attenuation loss proposed in [30] to learn the uncertainty. As there is no 'uncertainty label', it is implicitly learned from the loss function.

An important contribution in this paper is a novel *Probabilistic DMT* modeling the intra-structural uncertainty, due to the geometry, topology, as well as the segmentation model's confidence. The classic DMT cannot capture the inherent uncertainty of a structure; it produces a single representation for the structure, i.e., a one-pixel-wide skeleton. However, this skeleton is computed in a deterministic manner. Such deterministic computation is rigid and fails to capture possible variations of a structure.

Instead, our Probabilistic DMT (Prob. DMT) represents each structure as sample skeletons drawn from an underlying generative process guided by the likelihood map (the skeleton resulting from the

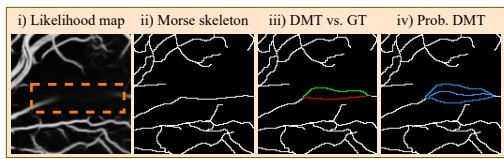 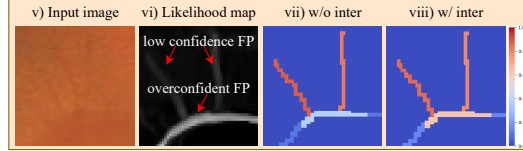

| i) Likelihood map | ii) Morse skeleton | iii) DMT vs. GT | iv) Prob. DMT |

a) Intra-structural uncertainty

| v) Input image | vi) Likelihood map | vii) w/o inter | viii) w/ inter |

b) Inter-structural uncertainty

Figure 2: (a) Intra-structural uncertainty: In the likelihood map (i), we highlight a false negative (FN) structure missed by the segmentation network. In (ii), we show the skeletons representing structures from classic DMT. In (iii), we highlight the GT (green) and the incorrect DMT skeleton (red) for the FN structure. In (iv), we show skeleton samples by Prob. DMT (blue); (b) Inter-structural uncertainty (*inter*): a retinal image with very weak signal (v). The likelihood map (vi) shows three potential false positive (*FP*) structures, two vertical and one horizontal. Without inter-structural uncertainty (vii), the horizontal structure has high confidence. With inter-structural uncertainty (viii), the horizontal structure gets higher uncertainty influenced by the two vertical structures.

original DMT being one of the samples). As illustrated in Fig. 2(iii), the original DMT generates a skeleton that significantly deviates from the true structure due to the uncertainty inherent in the likelihood. In contrast, Prob. DMT effectively captures potential variations (Fig. 2(iv)), offering valuable insights into the impact of uncertainty on the structural composition. The greater the variation, the greater the intra-structural uncertainty. During training, we repeatedly sample skeletons for each structure, and feed the samples to the joint inference model for uncertainty prediction. Indeed, as shown in Fig. 2(iv), sampling multiple skeletons also leads to a better chance of uncovering the true structure. This observation inspires us to use the uncertainty estimation method to re-calibrate the original segmentation model and achieve even better segmentation performance.

We note that the method in [28] (which we refer to as Hu et al.) also used DMT to decompose structures and estimate their uncertainty. However, this method used the classic DMT to deterministically generate skeletons, and thus failed to model intra-structural uncertainty. Furthermore, instead of joint inference, the method sampled from all configurations; and to reduce the computational burden, it pruned the exponential-sized configuration space using a saliency measure called persistence [9, 64, 71]. The pruning was very coarse, thus resulting in suboptimal uncertainty estimation. As illustrated in Fig. 1(e), Hu et al. produces overconfident maps; most structures, including many false negatives and false positives, are assigned zero uncertainty. In contrast, our method produces much better uncertainty estimates (Fig. 1(f)), owing to the proper modeling of both intra-structural and inter-structural uncertainties.

We summarize the main contributions of this paper as follows:

1. We propose a novel method to estimate the uncertainty of a given segmentation network at *a structural level*.
2. We propose Probabilistic DMT, a probabilistic method to generate structural variations and to capture intra-structural uncertainty.
3. We propose a joint prediction model on all the structures in order to capture inter-structural uncertainty.

Empirical evaluation shows that our method achieves much better uncertainty estimates on both 2D and 3D datasets, outperforming existing methods.

## 2   Related work

**Topology-guided image segmentation.** Several works focus on maintaining the correct connectivity or topology of thin structures. Topology-aware loss functions [46, 62, 7, 27, 72, 22, 26] impose per-pixel constraints to improve topological integrity. Discrete Morse theory has also been used to improve the topological awareness of segmentation networks [29, 9, 11, 56, 71, 3]. These approaches use topological tools to improve segmentation at a pixel level, which is a weaker constraint compared to the structural level. In contrast, our method performs joint reasoning directly over the structures.

**Uncertainty quantification.** In recent years, there has been significant work on uncertainty quantification (UQ) of deep neural networks [1, 20, 36]. Here we review UQ techniques tailored for semantic segmentation. *Pixel-wise uncertainty:* Semantic segmentation is a per-pixel classification

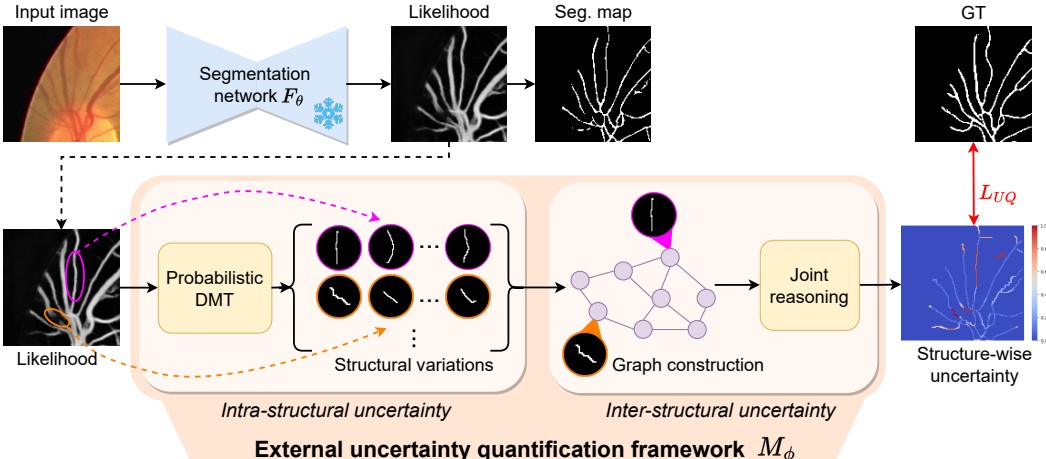

Figure 3: An overview of the proposed method $M_\phi$. The given segmentation network $F_\theta$ has frozen weights. Probabilistic DMT decomposes the likelihood into structures, and samples skeleton representations of each. A graph is then constructed over the structures to perform joint reasoning of their uncertainty. The training is supervised by comparing with the GT (via the loss $L_{UQ}$, red arrow).

task and naturally most UQ methods produce per-pixel uncertainty estimates. In [30], the authors propose a Bayesian framework using MC dropout [16] and a learned loss attenuation to respectively capture model and data uncertainty. Recent methods have turned to generative models to generate multiple hypotheses, and the per-pixel variance across the hypotheses is treated as uncertainty. Some works in this direction are an ensemble of $M$ networks [34], a single network with $M$ heads [58], Prob.-UNet [32], and PHiSeg [4]. Prob.-UNet integrates a conditional variational autoencoder [63] with UNet [57], generating multiple hypotheses via latent variable sampling. PHiSeg extends this by introducing latent variables at every UNet level, thereby producing more diverse samples. *Structure-wise uncertainty:* Methods such as [41, 60] compute structure (volume) uncertainty by averaging over the pixel-wise uncertainty estimates. The method closest to ours is Hu et al. [28]. It is a generative model derived from Prob.-UNet where the latent variable has meaning in topology (specifically, a global persistence threshold). This threshold severely limits the structure space, overlooking several false positive/negative structures. Thus they tend to produce overconfident uncertainty estimates.

## 3 Method

Given a trained segmentation network, our goal is to capture the uncertainty of its prediction at a structural level. Note that we do not modify the network in any way; instead, we propose an external module that reasons the uncertainty of each structure in the segmentation. Fig. 3 provides an overview of our method. Let $F_\theta$ denote the trained segmentation network, and $M_\phi$ denote our proposed external uncertainty quantification framework. $M_\phi$ takes as input the likelihood map of $F_\theta$ and the input image. It generates a set of structures, and estimates an uncertainty value for each of them. During training, $M_\phi$ is trained by comparing with the ground truth (GT) annotation.

$M_\phi$ consists of two primary modules to capture intra-structural and inter-structural uncertainty. The first module, Probabilistic DMT (Prob. DMT), generates structures based on the likelihood map. For each structure, it samples a set of skeletons representing different variations. Details are provided in Sec. 3.1. The second module jointly predicts the uncertainties of all the structures. At each training iteration, it takes one sample skeleton for each structure, plus the likelihood map and input image, as input. Details are described in Sec. 3.2. Throughout the sections, we consider one data sample $(x, y)$ where $x$ is an input image and $y$ is the segmentation GT. The likelihood map is $f = F_\theta(x)$.

### 3.1 Modelling the structural space

In this section, we first describe how DMT obtains the constituent structures of a likelihood map. Then we propose our Prob. DMT formulation to capture intra-structural uncertainty.

**Discrete Morse theory.** Consider the likelihood map $f$ generated from the segmentation network $F_\theta$. We wish to decompose $f$ into a set of structures, capturing not only the salient structures but also the faint ones. In the segmentation map, salient and faint structures broadly correspond to true positive and false negative structures. In Fig. 4(b), we highlight the false negative (FN) structures. These structures are missed in the segmentation, but will be captured by DMT (Fig. 4(d)).

DMT treats the likelihood map $f$ as a terrain function, decomposing it into a *Morse complex* consisting of critical points, paths connecting them, patches in between paths, and volumes enclosed by patches (for 3D images). *Critical points* are locations $w$ with zero gradients ($\nabla f(w) = 0$), i.e., minima, maxima, or saddle points. Paths, called *V-paths*, are routes connecting critical points via the non-critical ones. A V-path connecting a saddle point to a maxima is called a stable manifold. These stable manifolds are the underlying terrain's mountain ridges, and delineate structures of interest. In Fig. 4(c), we show the locations of saddles and maximas in the Morse complex, and in Fig. 4(d), we show the union of all the stable manifolds connecting them. In this paper, we only focus on the zero- and one-dimensional Morse structures, i.e., the union of all stable manifolds and their associated saddle and maxima. We call the collection of such structures the Morse skeleton.

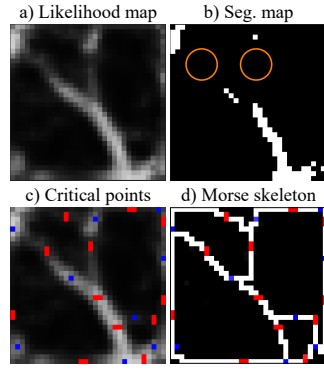

Figure 4: Orange indicates FN structures; (c) shows saddle points (red) and maximas (blue), and omits minimas; (d) shows the union of the stable manifolds of the saddle points.

By default, DMT generates stable manifolds in a completely deterministic manner, failing to take into account the intra-structural uncertainty in the likelihood $f$. Therefore, these stable manifolds may fail to correctly delineate the true structure, as shown in Fig. 5.

**Probabilistic DMT.** To account for the inherent uncertainty, we explicitly model the structure as a collection of samples from an underlying generative process. The skeleton from the original DMT is just one possibility out of many. The method is achieved via a perturb-and-walk algorithm, in which we iteratively perturb the likelihood map, and regenerate the skeleton.

The rationale is that the likelihood map is a weighted aggregation of all possible skeleton representations. To inverse the aggregation and recover these skeletons is challenging. Instead, we follow the classic *perturb-and-map principle*, which was used to efficiently sample from a complex discrete graphical model distribution [52, 24, 35]. We randomly perturb the likelihood function. For each perturbed likelihood, we compute a skeleton as a sample. See Fig. 5 for an illustration. The sampled skeletons will reflect the uncertainty properly. For a structure that is less salient in the likelihood map, the sample skeletons will have large variations, generating a large uncertainty. For a salient structure in the likelihood map, the sample skeletons will be less variant, resulting in a low uncertainty.

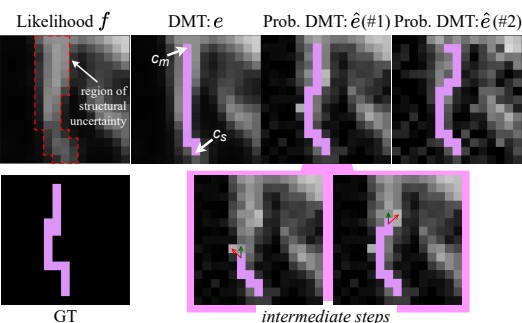

Figure 5: Structures (#1,#2) sampled from the distribution. Green arrow is path chosen using $Q(c')$; red arrow is next step w/o considering $Q_d(c')$.

Assume a given likelihood function $f$ and one of its structures, represented by a V-path $e$ connecting a saddle-maximum pair $(c_s, c_m)$. We generate a sample skeleton of the structure by first perturbing the likelihood with random noise. Next, we generate a path connecting $c_s$ and $c_m$. Recall in the original DMT, the skeleton is generated by following the mountain ridge. In other words, we start from the saddle point, and "walk" towards the maximum. At every step, we always walk to the neighboring pixel with the highest likelihood value. In Prob. DMT, we follow the same principle on the perturbed likelihood. However, the noisy perturbation of likelihood can cause the path to grow astray. Therefore, we additionally apply a distance-based regularizer to guide the walk towards the target $c_m$. We describe the process in detail below.

Let $e$ denote the structure obtained by following the V-path between $(c_s, c_m)$ in the original DMT. In order to generate its sample skeleton $\hat{e}$, we first draw a likelihood $f_n$ from a distribution centered on $f$

as $f_n \sim f + r$. This process is independent of the perturbation model $r$ used, and we use a Gaussian model in this work. As the variance of the Gaussian model is unknown, we use Bayesian probability theory to sample the variance from the Inverse Gamma distribution (its conjugate prior [49]).

Once we obtain $f_n$, we regenerate the path between $(c_s, c_m)$. We take inspiration from random walk [38] as well as probability regularized walk [47] to generate the variant structure $\hat{e}$ from $f_n$. Our walk algorithm continuously grows $\hat{e}$ starting from $c_s$ and ending at $c_m$, one pixel at a time. The algorithm considers both the terrain $f_n$ and the distance to the destination $c_m$ to ensure path completeness. During the walk, given the current pixel location $c$, the next location $c''$ is chosen as $c'' = \text{argmax}(Q(c'))$, where, $c' \in \text{neighborhood}(c)^2$ and $Q(c') = \gamma Q_d(c') + (1 - \gamma) f_n(c')$, and, $Q_d(c') = \frac{1}{\|c_m - c'\|_2}$. We begin with $c := c_s$ and continue in this manner $c := c''$ till we reach $c_m{}^3$. In Fig. 5, we show a deterministic structure obtained from DMT along with sample variations produced by our method. We demonstrate the intermediate steps in the algorithm: the red arrow denotes the next step without considering the distance regularizer $Q_d$, while the green arrow denotes the next step using our formulation $Q$. Notice how only considering $f_n$ without $Q_d$ can prevent the path from reaching $c_m$. We thus require $Q_d$ to guide the path to completeness.

The structure $\hat{e}$ is a different realization of $e$, making each run of the Prob. DMT a stochastic one. We are thus able to explicitly model the structures as samples from a probability distribution. We also note that DMT is a special case of Prob. DMT when $r = 0$ and $\gamma = 0$. In practice, with some probability, we consider the original structure $e$ from DMT over generating its variant $\hat{e}$. Specifically, following a Bernoulli distribution, with a small probability $u$ we retain $e$, while with probability $1 - u$ we sample its variant $\hat{e}$ using the perturb-and-walk algorithm outlined above. This process is done separately and in parallel for every structure. The structures taken together form a Morse skeleton. The output of Prob. DMT is effectively one sample skeleton from the space of Morse skeletons. We provide further information regarding Prob .DMT in the Appendix: A outlines the pseudocode, B discusses the hyperparameters, and C reports its computational complexity.

## 3.2  Joint estimation of structural uncertainty

The Prob. DMT module gives us a set $E$ of structures. Our final step is to jointly reason about the uncertainty of all of them. To achieve this, we use a regression network that takes as input each structure $e \in E$, and outputs whether it is a true positive and the uncertainty of $F_\theta$ in predicting it.

**Details of the network.** Structures interact with each other in the image space and are not independent. During uncertainty estimation, it is therefore crucial to consider their spatial context, i.e., inter-structural uncertainty. Hence, we use Graph Neural Networks (GNN) [59], specifically Graph Convolution Networks (GCN) [31], to jointly reason about the structures and capture the high-order spatial interactions. In the graph, each node represents a structure, and edges between nodes exist when corresponding structures have non-zero overlaps (typically at endpoints). The input feature vector for each node is constructed as shown in Fig. 6. For every structure, we first concatenate $[x^c, f^c, m]$, where $x^c$ comes from the original input $x$; $f^c$ from the likelihood map $f$ (not $f_n$); and $m$ is a binary map indicating the presence of the structure. These $x^c, f^c, m$ are smaller crops/bounding boxes centered on the structure. After passing them through convolution blocks, we apply channel-wise pooling to obtain a fixed-length feature vector for training. We further concatenate the persistence value of the saddle point associated with the original DMT structure (aka stable manifold). Persistence value (from persistent homology [13]) is defined as the difference of function (likelihood) values of 2 critical cells (saddle-maxima pair). It captures the importance of a structure, thus making it a valuable feature in our framework. Note that we do not use the perturbed $f_n$ from the Prob. DMT method when constructing the feature vector.

**Training the network.** We train the regression network using the attenuation loss proposed in [30]. As there are no labels to learn uncertainty, it is implicitly learned during regression optimization. We fix a Gaussian likelihood, and so variance $\hat{\delta}^2$ is used as a measure of uncertainty. The network's head is split into two — to predict $\hat{p}(e)$ of being a true positive structure and its associated uncertainty $\hat{\delta}_e^2$. For numerical stability, we actually predict the log variance $s_e = \log \hat{\delta}_e^2$. The training loss is given in Eq. 1. The structures that we obtain from Prob. DMT may not always fully overlap with the true GT structures, that is, some structures may only have partial overlap. We thus do not impose any hard

---

[2]For neighborhood, we use 8-connectivity for 2D, and 26-connectivity for 3D in this work.

[3]If the path does not reach $c_m$, we impose a maximum limit on the update steps to prevent an infinite loop.

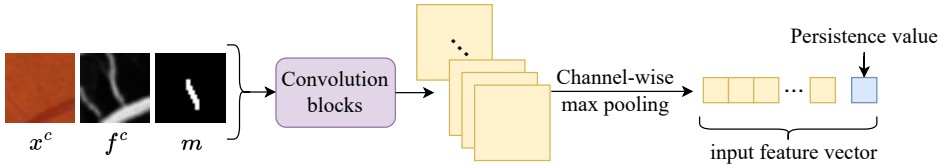

Figure 6: Construction of the input feature vector for each node (structure) in the GCN.

constraints in Eq. 1, instead, $z_e$ is a soft label, and is given by: $z_e = (\sum y \odot m)/(\sum m)$, where $y$ is the GT and $\odot$ is the Hadamard product. This value simply represents the proportion of the structure that overlaps with the GT, i.e., the fraction of the structure that is a true positive.

$$L_{UQ}(\phi) = \frac{1}{|E|} \sum_{\forall e \in E} \left( \frac{1}{2} \frac{\|\hat{p}(e) - z_e\|^2}{\exp(s_e)} + \frac{1}{2} s_e \right) \tag{1}$$

In [30], $\hat{\delta}^2$ denoted the pixel-wise uncertainty of the framework's input. In our setting, the input to our framework is $f = F_\theta(x)$, and so $\hat{\delta}^2$ is modeled to capture the structure-wise uncertainty inherent in data $x$ and model $F_\theta$. Training $\hat{\delta}^2$ in this manner ensures that the network does not trivially predict high or low uncertainty, rather, predicts an uncertainty estimate that is dependent on the input.

### 3.3 Proposed module $M_\phi$

For Eq. 1 to hold, we require $M_\phi$ to be a probabilistic network. We already show in Sec. 3.1 our formulation for Prob. DMT. Additionally, the regression network is also probabilistic as we use MC dropout [16].

**Inference procedure.** We take $T$ runs of $M_\phi$ and compute the uncertainty as the mean $\bar{\delta}^2_e = \frac{1}{T} \sum_{t=1}^{T} (\hat{\delta}^2_e)_t$. We similarly obtain $\bar{p}(e)$ from $\hat{p}(e)$. In Fig. 7, we illustrate the post-processing steps to obtain the structure-wise uncertainty heatmap. First, we obtain maps $\bar{p} = \cup \bar{p}_e$ and $\bar{\delta}^2 = \cup \bar{\delta}^2_e$ having the same spatial resolution as the input $x$. We then binarize $\bar{p}$, and overlay it onto the segmentation map obtained from $F_\theta$. We do this because Prob. DMT

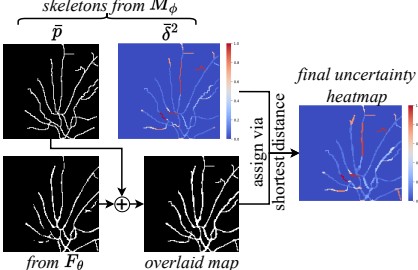

Figure 7: Post-processing procedure.

gives us one-pixel wide skeleton structures but we need to recover the structure thickness. Next, we use shortest distance to assign uncertainty values from $\bar{\delta}^2$ to the pixels in the overlaid map. The shortest distance uses paths only along the foreground pixels. In Fig. 7 we show how we obtain the final uncertainty heatmap from the skeleton heatmap. We also note that the overlaid map is an additional output of our method: it is an improved segmentation map that can be used instead of the one obtained by $F_\theta$. We provide more details in Appendix D.

## 4 Experiments

**Datasets.** We evaluate our method on four datasets: **DRIVE** [65], **ROSE** [40], **ROADS** [44] and **PARSE 2022 Grand Challenge** [39, 70]. The DRIVE dataset contains 2D retinal vasculature; ROSE is a 2D retinal OCTA (Optical Coherence Tomography Angiography) segmentation dataset, ROADS is a large non-medical dataset containing aerial images, and PARSE contains 3D CT scans of pulmonary arteries. We further describe the datasets and the data splits in Appendix E.

**Baselines.** We broadly split our comparison baselines into three types: a) Standard vessel segmentation methods: **UNet** [57], **DeepVesselNet** [68], and **CS$^2$-Net** [48]; b) Pixel-wise uncertainty estimation methods: **Prob.-UNet** [32], and **PHiSeg** [4]; c) Structure-wise uncertainty estimation method: **Hu et al.** [28]. Implementation details are provided in Appendix F.

**Evaluation metrics.** To evaluate the quality of uncertainty quantification, we use **Expected Calibration Error (ECE)** [50] and **Reliability Diagrams (RD)** [8]. Furthermore, we also evaluate the segmentation on metrics such as **DICE** [73], **clDice** [62], **ARI** [2], **VOI** [42], **Betti Number error** [27] and **Betti Matching error** [66]. We include clDice, ARI, VOI, Betti Number and Betti

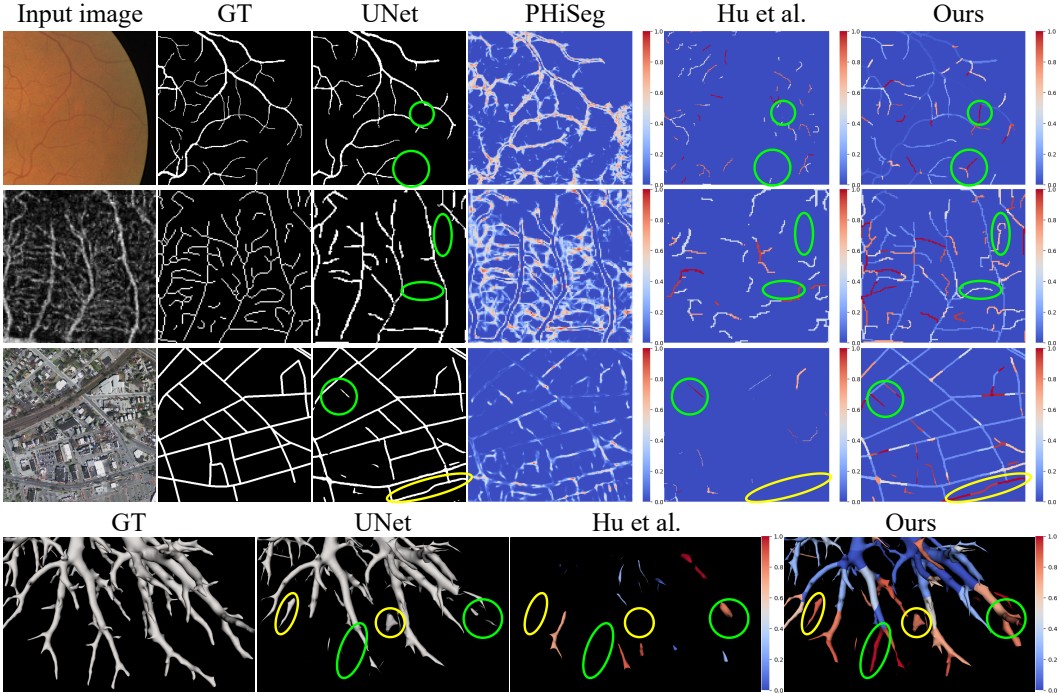

Figure 8: Qualitative results compared to the uncertainty baselines. We show uncertainty estimates in the form of a heatmap. Green highlights false negatives and yellow highlights false positives. Row 1: DRIVE; Row 2: ROSE; Row 3: ROADS; Row 4: PARSE (3D render).

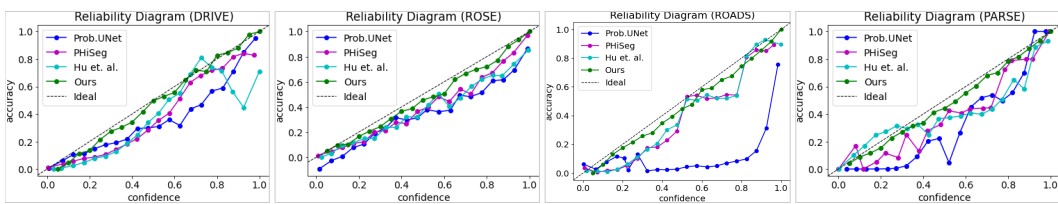

Figure 9: Reliability diagrams of samples from each dataset.

Matching as they are topology-based metrics and hence are sensitive to the performance on thin structures. Detailed definitions are present in Appendix G.

## 4.1 Results

Tab. 1 shows the quantitative results against uncertainty methods, and Tab. 2 shows the quantitative results on different backbone architectures. We show the respective qualitative results in Fig. 8 and Fig. 10. In Fig. 9, we plot the Reliability Diagrams. We also perform the unpaired **t-test** [67] (95% confidence interval) to determine the statistical significance. Each table reports the mean and standard deviations for every metric, with statistically significant better performances in bold and numerically better (but not significant) performances in italics. For all the probabilistic methods, the average of five runs was used. For our method, we generated the structure-wise uncertainty estimates and the segmentation map by following the steps outlined in the 'Inference procedure' in Sec. 3.3. Due to space constraints, results on two metrics Betti Number and Betti Matching are reported in Appendix H. We discuss the remaining performances below.

**Performance of uncertainty estimation.** Tab. 1 shows that our method outperforms others on both ECE and segmentation metrics. Fig. 9 displays RDs, with our method following the ideal line much closely compared to others. This is because we explicitly model the distribution of the structures, thereby quantifying the uncertainty of the segmentation network. In Fig. 8, we also see that our method generates better fidelity structure-wise uncertainty maps compared to Hu et al. Our heatmaps

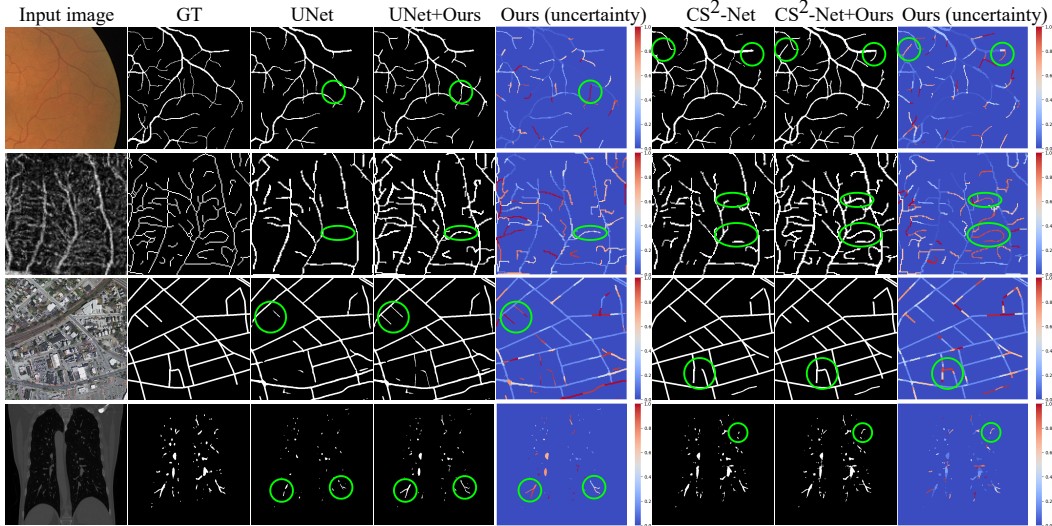

| Input image | GT | UNet | UNet+Ours | Ours (uncertainty) | CS²-Net | CS²-Net+Ours | Ours (uncertainty) |

Figure 10: Qualitative results over different segmentation backbones. Green highlights false negatives. Row 1: DRIVE; Row 2: ROSE; Row 3: ROADS; Row 4: PARSE.

Table 1: Comparison against uncertainty baselines (all use UNet [57] as the backbone)

| Dataset | Method | ECE (%)↓ | Dice↑ | clDice↑ | ARI↑ | VOI↓ |
|---|---|---|---|---|---|---|
| DRIVE | Prob.-UNet [32] | 8.3316 ± 0.0043 | 0.7779 ± 0.0219 | 0.7663 ± 0.0492 | 0.7759 ± 0.0532 | 0.3560 ± 0.0203 |
| | PHiSeg [4] | 7.9316 ± 0.0032 | 0.7851 ± 0.0295 | 0.7712 ± 0.0497 | 0.7767 ± 0.0497 | 0.3527 ± 0.0308 |
| | Hu et al. [28] | 8.0883 ± 0.0036 | 0.7866 ± 0.0141 | 0.7725 ± 0.0392 | 0.7768 ± 0.0403 | 0.3489 ± 0.0286 |
| | Ours | **4.1633 ± 0.0043** | **0.7976 ± 0.0195** | **0.7974 ± 0.0372** | **0.7996 ± 0.0301** | **0.3322 ± 0.0229** |
| ROSE | Prob.-UNet [32] | 7.2795 ± 0.0022 | 0.7378 ± 0.0284 | 0.6485 ± 0.0258 | 0.7219 ± 0.0538 | 0.7769 ± 0.0146 |
| | PHiSeg [4] | 7.0875 ± 0.0036 | 0.7415 ± 0.0267 | 0.6552 ± 0.0236 | 0.7309 ± 0.0425 | 0.7638 ± 0.0128 |
| | Hu et al. [28] | 6.9243 ± 0.0033 | 0.7429 ± 0.0132 | 0.6598 ± 0.0172 | 0.7506 ± 0.0302 | 0.7616 ± 0.0123 |
| | Ours | **3.9904 ± 0.0041** | **0.7593 ± 0.0171** | **0.6782 ± 0.0119** | **0.7837 ± 0.0314** | **0.7403 ± 0.0239** |
| ROADS | Prob.-UNet [32] | 8.4318 ± 0.0042 | 0.7194 ± 0.0418 | 0.8058 ± 0.0615 | 0.7350 ± 0.0494 | 0.5602 ± 0.0308 |
| | PHiSeg [4] | 7.9331 ± 0.0038 | 0.7203 ± 0.0366 | 0.8113 ± 0.0521 | 0.7392 ± 0.0416 | 0.5559 ± 0.0295 |
| | Hu et al. [28] | 7.8034 ± 0.0029 | 0.7275 ± 0.0361 | 0.8282 ± 0.0493 | 0.7314 ± 0.0391 | 0.5644 ± 0.0239 |
| | Ours | **4.1442 ± 0.0031** | **0.7461 ± 0.0364** | **0.8496 ± 0.0455** | **0.7601 ± 0.0349** | **0.5463 ± 0.0218** |
| PARSE | Prob.-UNet [32] | 9.9918 ± 0.0069 | 0.6002 ± 5.7751 | 0.6179 ± 0.0804 | 0.6523 ± 0.0654 | 0.8923 ± 0.0417 |
| | PHiSeg [4] | 9.9280 ± 0.0077 | 0.5910 ± 3.0858 | 0.6080 ± 0.0743 | 0.6512 ± 0.0521 | 0.8839 ± 0.0297 |
| | Hu et al. [28] | 7.7891 ± 0.0075 | 0.6044 ± 2.3583 | 0.6153 ± 0.0724 | 0.6537 ± 0.0363 | 0.8803 ± 0.0318 |
| | Ours | **4.0289 ± 0.0073** | *0.6190 ± 3.0826* | **0.6221 ± 0.0613** | **0.6658 ± 0.0461** | **0.8701 ± 0.0332** |

assign non-zero uncertainty to several false positives/negatives in the backbone UNet's outputs. This is because we reason about every structure while Hu et al. limits the structure space via pruning.

**Performance over different backbones.** Tab. 2 and Fig. 10 show that our method is backbone-agnostic. It consistently improves the segmentation quality and produces high fidelity uncertainty maps for each of the underlying networks. This validates the practical applicability of our method.

**Performance of proofreading.** One of the motivations of this work is to streamline the proofreading process. Structure-wise uncertainty can be used as a guide, with a user having to simply accept/reject a structure. We conduct experiments on the ROSE dataset and simulate user interaction with our method and Hu et al.'s. The user is given each method's final segmentation map, and inspects structures in decreasing order of uncertainty (till 0.5). Each uncertain structure is subjected to a yes/no decision, which is denoted as one 'click'. The results are in Fig. 11. Our findings are consistent with the observation that Hu et al. assigns zero uncertainty to many structures; thus their margin of improvement is limited and saturates quickly.

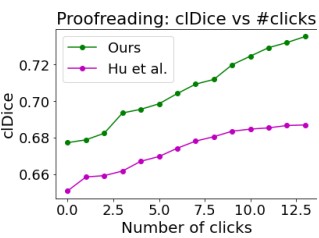

Figure 11: Proofreading.

Table 2: Comparison against different segmentation backbones

| Dataset | Method | Dice↑ | clDice↑ | ARI↑ | VOI↓ |
|---|---|---|---|---|---|
| DRIVE | UNet [57] | $0.7728 \pm 0.0336$ | $0.7586 \pm 0.0405$ | $0.7530 \pm 0.0519$ | $0.3697 \pm 0.0329$ |
| | UNet [57] + Ours | $\mathbf{0.7976 \pm 0.0195}$ | $\mathbf{0.7974 \pm 0.0372}$ | $\mathbf{0.7996 \pm 0.0301}$ | $\mathbf{0.3322 \pm 0.0229}$ |
| | DeepVesselNet [68] | $0.8015 \pm 0.0260$ | $0.7997 \pm 0.0431$ | $0.7729 \pm 0.0457$ | $0.3413 \pm 0.0256$ |
| | DeepVesselNet [68] + Ours | $\mathbf{0.8173 \pm 0.0190}$ | $\mathbf{0.8285 \pm 0.0361}$ | $\mathbf{0.8037 \pm 0.0361}$ | $\mathbf{0.3238 \pm 0.0192}$ |
| | CS$^2$-Net [48] | $0.8189 \pm 0.0176$ | $0.8125 \pm 0.0413$ | $0.8204 \pm 0.0495$ | $0.3417 \pm 0.0203$ |
| | CS$^2$-Net [48] + Ours | $\mathbf{0.8301 \pm 0.0172}$ | $\mathbf{0.8367 \pm 0.0305}$ | $\mathbf{0.8495 \pm 0.0301}$ | $0.3243 \pm 0.0258$ |
| ROSE | UNet [57] | $0.7375 \pm 0.0197$ | $0.6453 \pm 0.0165$ | $0.7206 \pm 0.0426$ | $0.8488 \pm 0.0126$ |
| | UNet [57] + Ours | $\mathbf{0.7593 \pm 0.0171}$ | $\mathbf{0.6782 \pm 0.0119}$ | $\mathbf{0.7837 \pm 0.0314}$ | $\mathbf{0.7403 \pm 0.0239}$ |
| | DeepVesselNet [68] | $0.7653 \pm 0.0101$ | $0.6634 \pm 0.0192$ | $0.7622 \pm 0.0302$ | $0.7426 \pm 0.0163$ |
| | DeepVesselNet [68] + Ours | $\mathbf{0.7795 \pm 0.0205}$ | $\mathbf{0.6873 \pm 0.0195}$ | $\mathbf{0.7936 \pm 0.0282}$ | $\mathbf{0.7164 \pm 0.0226}$ |
| | CS$^2$-Net [48] | $0.7623 \pm 0.0285$ | $0.6799 \pm 0.0127$ | $0.7702 \pm 0.0322$ | $0.7236 \pm 0.0157$ |
| | CS$^2$-Net [48] + Ours | $\mathbf{0.7886 \pm 0.0208}$ | $\mathbf{0.6968 \pm 0.0149}$ | $\mathbf{0.7981 \pm 0.0211}$ | $\mathbf{0.7072 \pm 0.0168}$ |
| ROADS | UNet [57] | $0.7011 \pm 0.0426$ | $0.7918 \pm 0.0679$ | $0.7143 \pm 0.0526$ | $0.5832 \pm 0.0345$ |
| | UNet [57] + Ours | $\mathbf{0.7461 \pm 0.0364}$ | $\mathbf{0.8496 \pm 0.0455}$ | $\mathbf{0.7601 \pm 0.0349}$ | $\mathbf{0.5463 \pm 0.0218}$ |
| | DeepVesselNet [68] | $0.7518 \pm 0.0345$ | $0.8248 \pm 0.0574$ | $0.7923 \pm 0.0441$ | $0.5641 \pm 0.0321$ |
| | DeepVesselNet [68] + Ours | $\mathbf{0.7673 \pm 0.0324}$ | $\mathbf{0.8513 \pm 0.0519}$ | $\mathbf{0.8139 \pm 0.0464}$ | $\mathbf{0.5357 \pm 0.0329}$ |
| | CS$^2$-Net [48] | $0.7539 \pm 0.0366$ | $0.8341 \pm 0.0511$ | $0.8197 \pm 0.0426$ | $0.5475 \pm 0.0468$ |
| | CS$^2$-Net [48] + Ours | $\mathbf{0.7692 \pm 0.0372}$ | $\mathbf{0.8559 \pm 0.0528}$ | $\mathbf{0.8368 \pm 0.0419}$ | $\mathbf{0.5261 \pm 0.0411}$ |
| PARSE | UNet [57] | $0.5905 \pm 3.0661$ | $0.6104 \pm 0.0727$ | $0.6509 \pm 0.0852$ | $1.9738 \pm 0.0414$ |
| | UNet [57] + Ours | $\mathit{0.6190 \pm 3.0826}$ | $\mathbf{0.6221 \pm 0.0613}$ | $\mathbf{0.6658 \pm 0.0461}$ | $\mathbf{0.8701 \pm 0.0332}$ |
| | DeepVesselNet [68] | $0.7208 \pm 3.0452$ | $0.6801 \pm 0.0554$ | $0.6923 \pm 0.0524$ | $0.4907 \pm 0.0701$ |
| | DeepVesselNet [68] + Ours | $\mathit{0.7376 \pm 3.1863}$ | $\mathbf{0.6983 \pm 0.0622}$ | $\mathbf{0.7098 \pm 0.0613}$ | $\mathbf{0.4711 \pm 0.0613}$ |
| | CS$^2$-Net [48] | $0.7630 \pm 3.9415$ | $0.6918 \pm 0.0695$ | $0.7138 \pm 0.0695$ | $0.4273 \pm 0.0521$ |
| | CS$^2$-Net [48] + Ours | $\mathit{0.7720 \pm 2.8109}$ | $\mathbf{0.7113 \pm 0.0689}$ | $\mathbf{0.7343 \pm 0.0733}$ | $\mathbf{0.4078 \pm 0.0642}$ |

## 4.2 Ablation studies

To demonstrate the efficacy of the proposed method, we conduct ablation studies of the different components in our pipeline, as well as check the effect of changing hyperparameter values. We also include ablation studies on the dimensionality of the input feature vector, and size of the crops/bounding boxes (which we report in Appendix I). All analyses are on the DRIVE dataset using UNet [57] as the backbone.

Table 3: Ablation of different modules

| DMT | Reg. Net | ECE (%)↓ | clDice↑ |
|---|---|---|---|
| DMT | GNN | $6.3481 \pm 0.0082$ | $0.7729 \pm 0.0304$ |
| Prob. DMT | MLP | $4.8202 \pm 0.0046$ | $0.7745 \pm 0.0305$ |
| Prob. DMT | GNN | $\mathbf{4.1633 \pm 0.0043}$ | $\mathbf{0.7974 \pm 0.0372}$ |

**Ablation of different modules.** We conduct ablation studies on both parts of our framework: structure generation (DMT vs Prob. DMT), and regression network (GNN vs Multi-layer Perceptron (MLP)). The results are in Tab. 3. Prob. DMT results in a sharp improvement in ECE compared to the original DMT; this supports our assertion that Prob. DMT models intra-structural uncertainty. Similarly, using GNN over MLP results in improvement. The message-passing in GNNs accounts for inter-structural uncertainty, thus yielding higher fidelity uncertainty estimates.

**Effect of hyperparameters.** Our main hyperparameters are $u, \alpha, \beta, \gamma$, with $u$ used in the Bernoulli distribution, $\gamma$ in the path-generation algorithm, and (shape $\alpha$, scale $\beta$) as prior hyperparameters of the Inverse Gamma distribution. We achieve the best ECE when $u = 0.3, \gamma = 0.2, \alpha = 2.0, \beta = 0.01$, however, a reasonable range always yields improvement, thus demonstrating the robustness of the method. We provide results of testing different hyperparameter values in Appendix I.

## 5 Conclusion

In this work, we propose to quantify the structure-wise uncertainty of a given segmentation network. Our framework explicitly models structures as samples from a probability distribution, thus helping to estimate intra-structural uncertainty. Furthermore, we incorporate inter-structural uncertainty by jointly reasoning over the structures, resulting in better fidelity uncertainty estimates. This structure-wise uncertainty quantification can streamline the proofreading process by reducing the time spent finding and correcting errors. Extensive experiments show the practical applicability of our method over different segmentation backbones and datasets. We further discuss the broader impact of our work and the limitations in Appendix J and Appendix K respectively.

**Acknowledgements.** This research was partially supported by grants NSF 2144901, NCI R21CA258493, and NIH 1R21CA258493-01A1. The content is solely the responsibility of the authors and does not necessarily represent the official views of the National Institutes of Health.

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

# Appendix

Appendix A provides a pseudocode of our proposed Probabilistic DMT module.

Appendix B provides a detailed discussion on the hyperparameters used in this work.

Appendix C reports the computational complexity of the method.

Appendix D expands on the 'Inference procedure' outlined in Sec. 3.3 of the main paper.

Appendix E provides the details of the datasets used.

Appendix F gives implementation details of our method and the baseline methods.

Appendix G provides definitions of the evaluation metrics used.

Appendix H provides results on topological metrics.

Appendix I provides results of ablation studies mentioned in Sec. 4.2 of the main paper.

Appendix J discusses the broader impact of this work.

Appendix K discusses the limitations of this work.

## A  Probabilistic DMT

In Algo. 1, we provide a pseudocode of our Probabilistic DMT module proposed in Sec. 3.1. We set $max\_step$ as 50 in our implementation. The terminologies used are: $f^c$ denotes the likelihood map from $F_\theta$ centered on structure $e$; $(c_s, c_m)$ are critical points between which the path $e$ is generated: $c_s$ denotes the saddle point and $c_m$ denotes the maxima. $IG$ denotes the Inverse Gamma distribution, and $\mathcal{N}$ denotes the Gaussian distribution.

---

**Algorithm 1** Probabilistic DMT pseudocode

---

1: **procedure** PROB_DMT($f^c, e, c_s, c_m$)
2:     $\hat{u} \sim$ Bernoulli($u$)
3:     **if** $\hat{u}$ is True **then**
4:         $\hat{e} \leftarrow e$
5:     **else**
6:         $\hat{e} \leftarrow$ GENERATE_PATH($f^c, c_s, c_m$)
7:     **end if**
8:     **return** $\hat{e}$
9: **end procedure**
10: **procedure** GENERATE_PATH($f, c_s, c_m$)
11:     **initialize** $m \leftarrow 0$                               ▷ $m$ has same spatial dimension as $f$
12:     **initialize** $c \leftarrow c_s$
13:     **initialize** $m[c] \leftarrow 1$
14:     **initialize** $step \leftarrow 0$
15:     $\sigma^2 \sim IG(\alpha, \beta)$
16:     $f_n \sim f + \mathcal{N}(0, \sigma)$
17:     **while** $c \neq c_m$ and $step < max\_step$ **do**
18:         $val \leftarrow 0$
19:         **for** $c' \in$ Neighborhood($c$) **do**
20:             $val[c'] \leftarrow \gamma * \frac{1}{\|c_m - c'\|_2} + (1 - \gamma) * f_n[c']$
21:         **end for**
22:         $c \leftarrow \text{argmax}(val)$                         ▷ Update current step
23:         $m[c] \leftarrow 1$
24:         $step \leftarrow step + 1$
25:     **end while**
26:     **return** $m$
27: **end procedure**

---

# B  Discussion of hyperparameters

The main hyperparameters in this work are $u, \alpha, \beta, \gamma$. We describe the importance of each below:

- $u$: This is the parameter for the Bernoulli distribution. In our Prob. DMT module, for every structure, we have a choice to either retain the structure as obtained from DMT, or, generate a sample skeleton using the perturb-and-walk algorithm. We model this choice using the Bernoulli distribution. Essentially, in some runs we would like the original DMT structures to also interact with the others. Thus a low value of $u$ works best. We found $u = 0.3$ to give the best performance, that is, for every structure there is a 30% chance that it's DMT form is used and a 70% chance that a sample variant is used. We find that $0.15 \leq u \leq 0.3$ have comparable performance.

- $\gamma$: This hyperparameter is used in the weighted combination of distance $Q_d$ and likelihood $f_n$ to obtain $Q(c')$, which is used to determine the next pixel location. It maintains a tradeoff between the distance regularizer $Q_d$ and the perturbed likelihood $f_n$. The higher the value of $\gamma$, the greater the distance regularizer, and consequently the generated path will become closer to that of a straight line. This is not desirable, as a straight line would lose the original composition of the structure. Additionally, because of the perturbation in the likelihood, we do not want the path to go astray. To ensure path completeness, we require $\gamma$ to be non-zero. Through experiments, we obtain the best performance when $\gamma = 0.2$.

- $\alpha, \beta$: These are prior hyperparameters of the Inverse Gamma (IG) distribution. We perturb the likelihood using a Gaussian model. As the variance of the Gaussian model is unknown, we use Bayesian probability theory to sample the variance from the IG distribution (its conjugate prior). And so, $\alpha$ is the shape parameter and $\beta$ is the scale parameter of this IG distribution. Ideally we would like a small perturbation of the likelihood and not a strong one. This is because a strong perturbation would corrupt wholly and we would not be able to sample a reasonable skeleton. At the same time, the perturbation should not be too small, otherwise we will not obtain a significant variant. The mean of the IG distribution is $\frac{\beta}{\alpha-1}$ (when $\alpha > 1, \beta > 0$), which on average is the value of the sampled variance for the Gaussian distribution. We achieve the best performance when $\alpha = 2.0$ and $\beta = 0.01$. The resulting sampled variance for the Gaussian model thus generates reasonable perturbation.

# C  Computational complexity

We report the inference time for 5 runs on a $256 \times 256$ input image patch as follows: Prob.-UNet: $0.196$ sec; PHiSeg: $1.811$ sec; Hu et al.: $5.485$ sec; Ours: $7.433$ sec.

The module which takes the most time is the DMT / Prob.DMT computation. Presently, this is the most optimized version as we have implemented it as an external module in C++. We will work towards porting the code to run on GPU to bring down the runtime even more.

Following [10], the computational complexity of DMT is $O(n \log n)$, where $n$ is the number of pixels in the image. Since Prob. DMT additionally computes structure variants, the complexity is $O(n \log n + m)$ where $m$ is approximately the number of foreground pixels, and typically $m << n$ for curvilinear structure datasets. The linear term is added as we traverse each foreground pixel only once when generating the sample skeleton.

# D  Inference procedure

We illustrate the inference procedure in Fig. 12. There are two outcomes of our framework $M_\phi$, namely, the structure-wise uncertainty heatmap as well as an improved discrete segmentation map that can be used instead of the one obtained by $F_\theta$.

For each structure $e$, we obtain $\hat{p}(e)$ (the regression output) and $\hat{\delta}_e^2$ (the uncertainty) from $M_\phi$. We take $T$ runs of $M_\phi$ and then for each structure $e$, we compute the mean across $T$ runs as $\bar{\delta}_e^2 = \frac{1}{T} \sum_{t=1}^{T} (\hat{\delta}_e^2)_t$, and, $\bar{p}(e) = \frac{1}{T} \sum_{t=1}^{T} \hat{p}(e)_t$. Next, we consider only those structures $e$ for which $\bar{p}(e) \geq 0.5$, i.e., $e$ has a minimum probability of $50\%$ of being a true positive. This is the *threshold* step in Fig. 12. We do this so as to consider only the true positive, false positive, and false negative

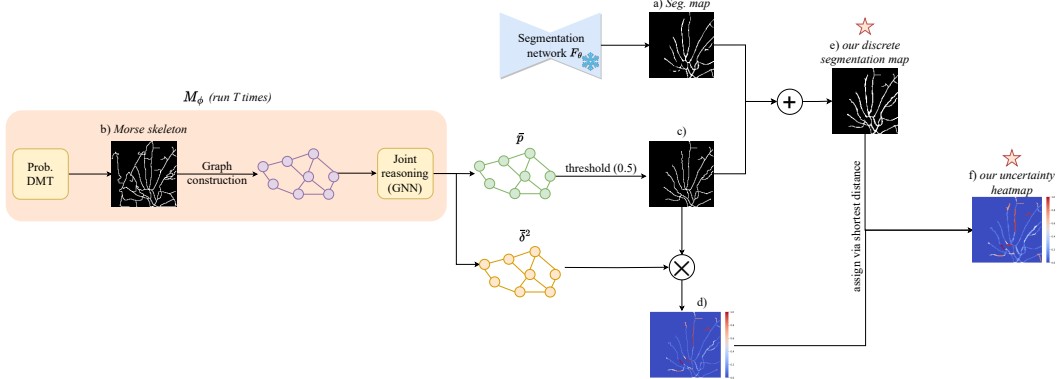

Figure 12: Inference procedure. Stars ($\star$) denote the final outcomes of our framework $M_\phi$.

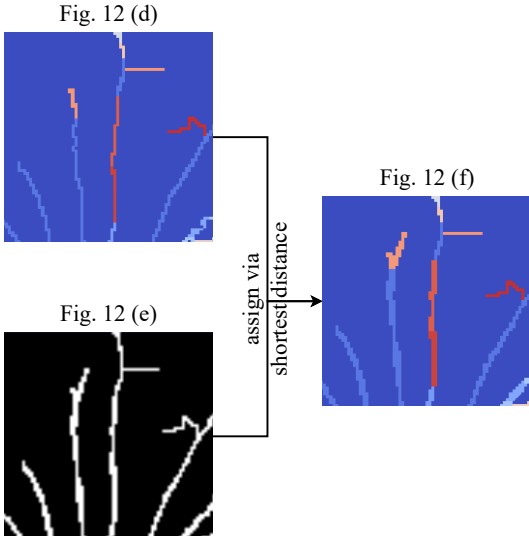

Figure 13: Zoomed-in views of Fig. 12.

structures in the final outcomes. We use these structures to create a skeletal discrete segmentation map (see Fig. 12(c)) which has the same spatial resolution as Fig. 12(a). As we want to recover the thickness of each structure, we overlay the two maps to get the final discrete segmentation map (see Fig. 12(e)).

The uncertainty heatmap that we obtain from $M_\phi$ is also skeletal (see Fig. 12(d)). We recover the structure thickness to get the final uncertainty heatmap (see Fig. 12(f)). We use shortest distance to do this. Shortest distance is used to assign uncertainty values from Fig. 12(d) to the pixels in Fig. 12(e). The shortest distance uses paths only along the foreground pixels and not along the background ones. This ensures that pixels within a structure are not assigned uncertainty values from other nearby structures. We provide a zoomed-in view in Fig. 13.

We note that generating the Morse complex is computationally heavy, however, it needs to be computed only once across the $T$ runs. As described in Sec. 3.1, the sampled structures are between $(c_s, c_m)$, and so the Morse complex is generated only in the first run.

# E   Dataset details

In this paper, we validate our results on four segmentation datasets: DRIVE [65], ROSE [40], ROADS [44], and PARSE 2022 Grand Challenge [39, 70].

**DRIVE [65]** The DRIVE dataset is a 2D retinal vessel dataset with $40$ images. Each image has a resolution of $584 \times 565$. We use the dataset's predetermined split of $20$ training images and $20$ test images. For training, we keep aside four randomly-chosen samples as validation, and train on the remaining $16$ samples.

**ROSE [40]** The ROSE dataset is a 2D retinal OCTA (Optical Coherence Tomography Angiography) segmentation dataset. We use ROSE-1 (SVC) in this work. It has a predetermined split of $30$ train and $9$ test samples, with each sample having a resolution of $304 \times 304$. For training, we keep aside four randomly-chosen samples as validation, and train on the remaining $26$ samples.

**ROADS [44]** The ROADS dataset is a large, non-medical dataset containing $1171$ aerial images ($1108$ train, $14$ validation, and $49$ test), each of $1500 \times 1500$ resolution. It is a challenging dataset due to obstruction from nearby trees, shadows, varying texture/color of roads, road class imbalance, and so on.

**PARSE [39, 70]** The PARSE dataset is a 3D CT dataset containing pulmonary artery segmentations. The dataset contains $100$ volumes and their sizes vary from $512 \times 512 \times 228$ to $512 \times 512 \times 376$. As there is no predetermined train/test split, we use $4$-fold cross-validation and report the average performance.

# F Implementation details

We use the PyTorch framework, a single NVIDIA Tesla V100-SXM2 GPU (32G Memory), and a Dual Intel Xeon Silver 4216 CPU@2.1Ghz (16 cores) for all the experiments. The training hyperparameters for our method as well as the baselines are as tabulated in Tab. 4. Note that although PARSE is a 3D dataset, all the segmentation networks (backbones) $F_\theta$ are 2D, that is, the networks are trained on 2D slices of the dataset. This was done to maintain a fair comparison across all baselines, as methods such as Prob.-UNet and PHiSeg only had 2D implementations available.

## F.1 Baselines

We use the publicly available codes for the baselines:

- UNet [57]: `https://github.com/johschmidt42/PyTorch-2D-3D-UNet-Tutorial`
- Prob.-UNet [32]: `https://github.com/stefanknegt/Probabilistic-Unet-Pytorch`
- PHiSeg [4]: `https://github.com/gigantenbein/UNet-Zoo`
- Hu et al. [28]: `https://github.com/HuXiaoling/Structural_Uncertainty`
- DeepVesselNet [68]: `https://github.com/dhavalshah18/deepvesselnet`
- CS$^2$-Net [48]: `https://github.com/iMED-Lab/CS-Net`

## F.2 Our method

Our code is available at `https://github.com/Saumya-Gupta-26/struct-uncertainty`. For reproducibility, we provide the architecture details as follows. The 'Joint reasoning' module described in Sec. 3.2 of our framework is a Graph Neural Network (GNN) [59], specifically a Graph Convolution Network (GCN) [31]. As per Fig. 6, the input feature vector for each graph node is constructed by passing $[x^c, f^c, m]$ through the following architecture: $C(3, 24) \to ReLU \to D(0.2) \to C(3, 32) \to ReLU \to D(0.2) \to MaxPool \to Concat(pers)$, where, $C(a, b)$ denotes a convolution layer having kernel size $a$ and number of output channels $b$; $D(p)$ denotes a Dropout layer with probability $p$; $MaxPool$ denotes the adaptive maxpool layer[4] returning a $1 \times 1$ output for each channel; and $pers$ is a scalar value denoting the persistence of the structure. Furthermore, the bounding box size of $[x^c, f^c, m]$ is $32 \times 32$ centered at each structure for 2D (for 3D, it is $32 \times 32 \times 32$).

The aforementioned layers generate the input feature vector for each graph node. They are then passed through the GNN which contains the following layers: $GCN(32) \to ReLU \to D(0.2) \to$

---

[4] `https://pytorch.org/docs/stable/generated/torch.nn.AdaptiveMaxPool2d.html`

Table 4: Training configuration

| Dataset | Model | Patch Size; Batch Size | Learning rate (LR); Optimizer |
|---|---|---|---|
| DRIVE | UNet | $128 \times 128$; 8 | 1e-3 with LR scheduler[6]; Adam[7] with weight decay 3e-5 |
| DRIVE | DeepVesselNet | $128 \times 128$; 8 | 1e-3 with LR scheduler; Adam with weight decay 3e-5 |
| DRIVE | $CS^2$-Net | $128 \times 128$; 8 | 1e-3 with LR scheduler; Adam with weight decay 3e-5 |
| DRIVE | Prob.-UNet | $128 \times 128$; 8 | 1e-3; Adam with weight decay 0 |
| DRIVE | PhiSeg | $128 \times 128$; 8 | 1e-4 with LR scheduler; Adam with weight decay 1e-5 |
| DRIVE | Hu et al. | $128 \times 128$; 8 | 1e-3; Adam with weight decay 0 |
| DRIVE | Ours | $128 \times 128$; 8 | 1e-3; Adam with weight decay 0 |
| ROSE, ROADS | UNet | $128 \times 128$; 6 | 1e-3 with LR scheduler; Adam with weight decay 3e-5 |
| ROSE, ROADS | DeepVesselNet | $128 \times 128$; 6 | 1e-3 with LR scheduler; Adam with weight decay 3e-5 |
| ROSE, ROADS | $CS^2$-Net | $128 \times 128$; 6 | 1e-3 with LR scheduler; Adam with weight decay 3e-5 |
| ROSE, ROADS | Prob.-UNet | $128 \times 128$; 6 | 1e-3; Adam with weight decay 0 |
| ROSE, ROADS | PhiSeg | $128 \times 128$; 6 | 1e-4 with LR scheduler; Adam with weight decay 1e-5 |
| ROSE, ROADS | Hu et al. | $128 \times 128$; 6 | 1e-3; Adam with weight decay 0 |
| ROSE, ROADS | Ours | $128 \times 128$; 6 | 1e-3; Adam with weight decay 0 |
| PARSE | UNet | $128 \times 128$; 8 | 1e-3 with LR scheduler; Adam with weight decay 3e-5 |
| PARSE | DeepVesselNet | $128 \times 128$; 8 | 1e-3 with LR scheduler; Adam with weight decay 3e-5 |
| PARSE | $CS^2$-Net | $128 \times 128$; 8 | 1e-3 with LR scheduler; Adam with weight decay 3e-5 |
| PARSE | Prob.-UNet | $128 \times 128$; 8 | 1e-3; Adam with weight decay 0 |
| PARSE | PhiSeg | $128 \times 128$; 8 | 1e-4 with LR scheduler; Adam with weight decay 1e-5 |
| PARSE | Hu et al. | $128 \times 128$; 8 | 1e-3; Adam with weight decay 0 |
| PARSE | Ours | $128 \times 128$; 8 | 1e-3; Adam with weight decay 0 |

$GCN(64) \rightarrow ReLU \rightarrow D(0.2) \rightarrow GCN(32) \rightarrow ReLU \rightarrow D(0.2)$, where $GCN(a)$ denotes a GCNConv layer [5] having $a$ number of output channels. The output from this sequence of layers is then fed to two separate $GCN(1)$ layers to output the regression likelihood $\hat{p}(e)$ and the uncertainty $\hat{\delta}_e^2$. As per GNN fashion, the weights of the layers are shared across all the nodes.

# G    Evaluation metrics

We use both segmentation and uncertainty metrics to evaluate our method. We describe the metrics in detail below.

## G.1    Uncertainty metrics

We use Reliability diagrams [8] and Expected calibration error (ECE) [50] to evaluate the quality of uncertainty. As both the metrics were originally designed for classification, we adapt from the classification task to semantic segmentation by treating each pixel as an independent sample. For both metrics, we first divide the probability interval $[0, 1]$ into $N$ equal-sized probability intervals (each of size $\frac{1}{N}$). We use $N = 20$ bins in this work. We then calculate the *accuracy* and *confidence* of each bin.

**Reliability diagrams (RD) [8]** Reliability diagrams are a visual representation of model calibration by plotting the expected accuracy as a function of confidence ($confidence = 1 - uncertainty$). Perfect calibration corresponds to an identity function in the RD, i.e., the model is not over/under-confident. Consider the set of pixels/structures whose predicted probabilities fall into the bin $B_i$. The accuracy and confidence are given by:

$$acc(B_i) = \frac{1}{|B_i|} \sum_{\forall x \in B_i} \mathbf{1}\left(\hat{Y}(x) = Y(x)\right)$$

$$conf(B_i) = \frac{1}{|B_i|} \sum_{\forall x \in B_i} \hat{P}(x)$$

---

[5] https://pytorch-geometric.readthedocs.io/en/latest/generated/torch_geometric.nn.conv.GCNConv.html#torch_geometric.nn.conv.GCNConv

[6] https://pytorch.org/docs/stable/generated/torch.optim.lr_scheduler.ReduceLROnPlateau.html

[7] https://pytorch.org/docs/stable/generated/torch.optim.Adam.html

Table 5: Comparison on topological metrics Betti Number error and Betti Matching error for different datasets. All methods use UNet as the backbone. $\beta_0^{\text{err}}$, $\beta_1^{\text{err}}$, and $\beta_2^{\text{err}}$ denote Betti Number error in 0-dim, 1-dim, and 2-dim respectively. $\mu_0^{\text{err}}$ and $\mu_1^{\text{err}}$ denote Betti Matching error in 0-dim and 1-dim respectively. The statistically significant better performances are highlighted in bold

| Dataset | Method | $\beta_0^{\text{err}}\downarrow$ | $\beta_1^{\text{err}}\downarrow$ | $\beta_2^{\text{err}}\downarrow$ | $\mu_0^{\text{err}}\downarrow$ | $\mu_1^{\text{err}}\downarrow$ |
|---|---|---|---|---|---|---|
| DRIVE | UNet | $166.3154 \pm 12.1065$ | $9.3149 \pm 4.2062$ | – | $205.8312 \pm 12.4496$ | $28.9587 \pm 5.3766$ |
| | Prob.-UNet | $146.6373 \pm 11.0831$ | $7.8197 \pm 3.1980$ | – | $191.2790 \pm 11.2324$ | $24.7826 \pm 4.9684$ |
| | PHiSeg | $145.3777 \pm 12.4873$ | $7.1542 \pm 3.6436$ | – | $190.0528 \pm 10.3376$ | $24.0893 \pm 4.1123$ |
| | Hu et al. | $140.8317 \pm 11.5502$ | $6.3083 \pm 2.2372$ | – | $188.6573 \pm 10.2403$ | $23.7263 \pm 4.3402$ |
| | Ours | $\mathbf{127.4041 \pm 10.7344}$ | $\mathbf{4.6172 \pm 2.6586}$ | – | $\mathbf{161.4536 \pm 9.7017}$ | $\mathbf{20.6835 \pm 3.4121}$ |
| ROSE | UNet | $231.5081 \pm 15.5573$ | $9.8826 \pm 1.6486$ | – | $243.2775 \pm 16.9274$ | $14.8922 \pm 1.6793$ |
| | Prob.-UNet | $229.7987 \pm 15.4307$ | $9.0396 \pm 1.6315$ | – | $240.3295 \pm 16.8371$ | $14.0771 \pm 2.0375$ |
| | PHiSeg | $220.0644 \pm 14.6356$ | $7.8644 \pm 2.0692$ | – | $228.4348 \pm 17.8907$ | $11.0377 \pm 1.9442$ |
| | Hu et al. | $219.7530 \pm 15.8446$ | $7.6981 \pm 1.5677$ | – | $226.2989 \pm 16.1992$ | $10.6838 \pm 1.8091$ |
| | Ours | $\mathbf{203.5791 \pm 13.6467}$ | $\mathbf{5.0553 \pm 1.4734}$ | – | $\mathbf{210.1763 \pm 15.1485}$ | $\mathbf{8.6489 \pm 1.5646}$ |
| ROADS | UNet | $75.6666 \pm 8.1079$ | $25.5777 \pm 7.4432$ | – | $78.2291 \pm 9.5055$ | $30.5104 \pm 6.6921$ |
| | Prob.-UNet | $70.3564 \pm 7.5929$ | $24.3852 \pm 7.0812$ | – | $72.8129 \pm 9.1638$ | $29.8830 \pm 6.1977$ |
| | PHiSeg | $68.7237 \pm 7.9177$ | $24.1772 \pm 6.5982$ | – | $70.2788 \pm 8.2474$ | $28.9467 \pm 5.9164$ |
| | Hu et al. | $61.5167 \pm 6.1625$ | $23.5863 \pm 5.3985$ | – | $62.4951 \pm 7.7601$ | $26.2681 \pm 5.8736$ |
| | Ours | $\mathbf{45.6735 \pm 5.9286}$ | $\mathbf{17.2653 \pm 4.6162}$ | – | $\mathbf{47.1429 \pm 6.7905}$ | $\mathbf{23.1837 \pm 5.4451}$ |
| PARSE | UNet | $673.7016 \pm 23.9541$ | $79.5825 \pm 10.9693$ | $18.4316 \pm 2.9432$ | – | – |
| | Prob.-UNet | $620.1903 \pm 22.0012$ | $51.4995 \pm 8.4096$ | $16.7046 \pm 2.2419$ | – | – |
| | PHiSeg | $587.2137 \pm 22.6801$ | $45.9331 \pm 8.7251$ | $15.8529 \pm 3.0218$ | – | – |
| | Hu et al. | $555.9788 \pm 23.5735$ | $40.0707 \pm 8.2376$ | $13.9498 \pm 2.2883$ | – | – |
| | Ours | $\mathbf{520.4991 \pm 22.4327}$ | $\mathbf{33.0532 \pm 7.8453}$ | $\mathbf{10.3831 \pm 2.1035}$ | – | – |

where, $Y$ is the discrete segmentation ground truth (GT), and $\hat{Y}$ is the discrete segmentation map outputted by the model. In our method, $\hat{Y}$ is as shown in Fig. 12(c). Additionally, $\hat{P}$ is the pixel-wise probability (likelihood) outputted by the model, whereas in our case, it is the structure-wise uncertainty $\bar{\delta}^2$ (Fig.12(d)). For our method and Hu et al., the $x \in B_i$ denotes structures, while in the other methods, it denotes pixels.

**Expected calibration error (ECE) [50]** RDs are only a visual cue, and so we also use ECE: a scalar to summarize the calibration performance. RDs do not take into account the number of pixels/structures in each bin. Thus, to account for such variations of the number of samples in a bin, we use ECE. It is given by:

$$ECE = \sum_{i=1}^{N} \frac{|B_i|}{n} |acc(B_i) - conf(B_i)|$$

where $n = \sum_{i}^{N} |B_i|$ is the total number of pixels/structures. The difference between $acc$ and $conf$ for a given bin represents the calibration gap. When there is perfect calibration, ECE is zero.

The definition of $acc$ and $conf$ remains the same as defined for RDs.

### G.2 Segmentation metrics

**DICE [73]** DICE score is a popular metric which measures the area/volumetric overlap between the predicted and ground truth discrete masks. It overcomes the class imbalance problem in the pixel-wise accuracy metric by considering only the foreground classes for measuring the overlap. The higher the DICE, the better the segmentation.

**clDice [62]** clDice is derived from DICE, however, clDice uses the skeleton of the predictions. This makes it sensitive to the performance of thin structures like vessels which is important in curvilinear segmentation. The higher the value, the better the segmentation.

**ARI [54]** The Rand index computes similarity between two clusterings. This raw score is "adjusted for chance" to get ARI (Adjusted Rand Index) [2]. The ARI takes into account the fact that some agreement between the two clusterings can occur by chance, and it adjusts the Rand index to account for this possibility. The higher the value, the better the segmentation.

**VOI [42]** The VOI metric is defined as the sum of the conditional entropies between two segmentations. A lower VOI value indicates better segmentation.

Table 6: Additional ablation study results of our method on the DRIVE dataset using UNet as the backbone. The best results (as reported in Tab. 1 of the main paper) are in bold. In the table, *Feature vector* denotes the length of the input feature vector to the GCN, while *Bounding box* denotes the size of the crop/bounding box centered on each structure. These hyperparameters are discussed in detail in Appendix B. The +1 denotes the concatenation of the scalar persistence value

| Feature vector | ECE (%)↓ | clDice↑ |
|---|---|---|
| $16 + 1$ | $4.9621 \pm 0.0048$ | $0.7906 \pm 0.0394$ |
| $32 + 1$ | $\mathbf{4.1633 \pm 0.0043}$ | $\mathbf{0.7974 \pm 0.0372}$ |
| $64 + 1$ | $4.1667 \pm 0.0045$ | $0.7972 \pm 0.0367$ |
| **Bounding box** | **ECE (%)↓** | **clDice↑** |
| $16 \times 16$ | $5.1085 \pm 0.0047$ | $0.7842 \pm 0.0416$ |
| $32 \times 32$ | $\mathbf{4.1633 \pm 0.0043}$ | $\mathbf{0.7974 \pm 0.0372}$ |
| $64 \times 64$ | $4.1689 \pm 0.0042$ | $0.7921 \pm 0.0363$ |

**Betti Number error [27]** This metric computes the difference between the Betti numbers of the prediction and the ground truth. We separately compute this metric for 0-dimension and 1-dimension structures.

**Betti Matching error [66]** This metric is similar to Betti Number, however, in Betti Matching, the spatial location of the 0-dimension and 1-dimension components is also taken into account. Hence this metric is stricter compared to Betti Number error.

## H    Quantitative results on topological metrics

In Tab. 5, we provide results on two topologically important metrics, namely, Betti Number error [27] and Betti Matching error [66]. Our method consistently improves the segmentation result in terms of topology. This is consistent with our results in Tab. 1 of the main paper where our method outperforms the baselines on other topology-based metrics like clDice [62], ARI [2] and VOI [42]. Note that for the 3D PARSE dataset, we were unable to provide Betti Matching error results as its official implementation handles only 2D inputs.

## I    Ablation study

We check the effect of the different hyperparameters in our work by conducting experiments on the DRIVE dataset using UNet as the backbone. Our main hyperparameters are $u, \alpha, \beta, \gamma$, with $u$ used in the Bernoulli distribution, $\gamma$ in the path-generation algorithm, and $(\alpha, \beta)$[8] as prior hyperparameters of the Inverse Gamma distribution. We test different values and report the ECE (the lower the better) in Fig. 14. For all the experiments, we set $u = 0.3$. We achieve the best ECE when $\gamma = 0.2, \alpha = 2.0, \beta = 0.01$, however, a reasonable range always yields improvement (notice how non-zero $\gamma$ results in a sharp improvement). This demonstrates the robustness of our proposed method.

We also provide additional ablation studies in Tab. 6. We now include ablation studies on the dimensionality of the input feature vector, and size of the crops/bounding boxes, and report this in Tab. 6. We obtain the best results when the input feature vector size is 32 and the bounding box is $32 \times 32$. For lower values (16 and $16 \times 16$), the performance reduces, while for higher values (64 and $64 \times 64$) we did not observe any statistically significant improvement. Thus to maintain the tradeoff between complexity and performance, we respectively use 32 and $32 \times 32$ for these hyperparameters.

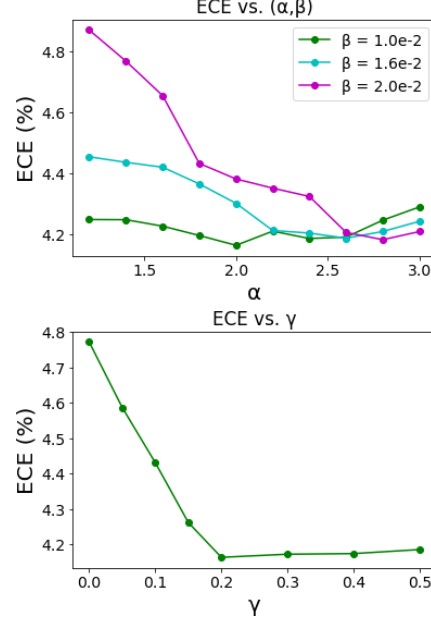

Figure 14: Effect of hyperparameters.

---

[8]$\alpha$ is the shape parameter; $\beta$ is the scale parameter.

## J  Broader impact

In this work, we aim to capture structure-wise uncertainty of a given network, where a structure is defined to be a coherent set of pixels a user can intuitively understand, e.g., small vessels/branches, short stretches of road etc. Fine-scale structures such as vessels, neurons, and membranes often consist of interconnected branches or structures that form a cohesive entity. Thus structure-wise uncertainty maps can highlight uncertain instances or branches as a whole, providing a more accurate indication of regions where the segmentation may be inaccurate or uncertain. This is beneficial for proofreading or error-correction tasks as they can direct the focus of human annotators to uncertain structures that require further attention. This can save time and effort compared to pixel-wise uncertainty maps that highlight numerous pixels as uncertain, many of which do not require correction. Thus structure-wise uncertainty can provide more interpretable estimates and is a desirable approach for improving segmentation accuracy and supporting downstream analysis tasks. This can go a long way as the benefit of proofreading is twofold: it improves segmentation quality, and it also helps expand the body of labeled data that can be further used to train automatic segmentation methods. Our work is thus a useful tool in streamlining the process of scalable annotation. At the present stage, we do not foresee any potential negative societal impacts.

## K  Limitations

Our method currently fits in the context of curvilinear segmentation. In general, large object segmentation could also benefit from structure-wise uncertainty (structures in this case would be smaller patches/volumes). Discrete Morse theory can be used in this setting, however, we would need to make use of topological features other than the stable manifold. In its present form, our proposed solution is currently not applicable in a setting beyond curvilinear segmentation.

