# OpenReview forum: "Topology-Aware Uncertainty for Image Segmentation"
_NeurIPS.cc/2023/Conference — NeurIPS 2023 poster_

### Official Review · Reviewer_ns8Y · 2023-07-02

**Soundness:** 3 good
**Presentation:** 2 fair
**Contribution:** 3 good
**Rating:** 5
**Confidence:** 3

**Summary:**

This paper proposes a framework that utilizes a probabilistic approach to extract structure-wise uncertainty estimates. This is achieved by extending the DMT to a probabilistic setting that models each structure as a sample from a probability distribution, thus capturing the intra-structural uncertainty. The proposed method then incorporates inter-structural uncertainty through a regression network which jointly reasons over the structures, using a Graph Neural Network (GNN). Additionally, a specialized inference procedure and post-processing steps are used to generate a structure-wise uncertainty heatmap, which can improve segmentation and quantify uncertainty more effectively.

**Strengths:**

- The authors demonstrate the versatility of their method by applying it to multiple segmentation network backbones and datasets. The proposed method was found to improve the quality of segmentation and produce high fidelity uncertainty maps for each network, making it backbone-agnostic.
- The introduction of a probabilistic DMT and a GNN to reason about inter-structural uncertainty is a significant advancement in this field.
- The proposed method for quantifying structure-wise uncertainty in segmentation networks has significant implications for medical image analysis. The ability to better understand and represent the uncertainty can potentially lead to improved segmentation quality, and could be crucial in medical applications where decision-making is based on these segmentation outputs.

**Weaknesses:**

- Small dataset size: DRIVE dataset 40 images. ROSE 39 images. PARSE 100 volumes. Larger dataset should be considered to evaluate the proposed method.
- Limited Explanation of Hyperparameter: while the authors have considered a range of hyperparameters in main body and supplementary, the paper could benefit from a more detailed discussion of the influence of these hyperparameters on the method's performance and how the optimal values were determined.

**Questions:**

Please add larger dataset and evaluate the proposed method. Also, explain more about the hyperparameters.

**Limitations:**

Please see the 'questions' part.

---

> ### Author Rebuttal · Authors · 2023-08-09
>
> Thank you for your constructive feedback! Please find our responses to specific queries below.
>
> **Q1:** Please add a larger dataset to evaluate the proposed method.
>
> **A1:** As recommended, we conduct these experiments. Please see 1. in the global response ‘Author Rebuttal by Authors’ above.
>
> **Q2:** Limited Explanation of Hyperparameter: while the authors have considered a range of hyperparameters in main body and supplementary, the paper could benefit from a more detailed discussion of the influence of these hyperparameters on the method's performance and how the optimal values were determined.
>
> **A2:** This is a good question and we provide a detailed discussion as requested.
>
> The main hyperparameters in this work are $u, \gamma, \alpha, \beta$. We describe the importance of each below:
> - $u$ : This is the parameter for the Bernoulli distribution, and we introduce it in L221 of the main paper. In our Prob. DMT module, for every structure, we have a choice to either retain the structure as obtained from DMT, or, generate a sample skeleton using the perturb-and-walk algorithm. We model this choice using the Bernoulli distribution. Essentially, in some runs we would like the original DMT structures to also interact with the others. Thus a low value of $u$ works best. We found $u = 0.3$ to give the best performance, that is, for every structure there is a 30% chance that it’s DMT form is used and a 70% chance that a sample variant is used. We find that $0.15 \leq u \leq 0.3$ have comparable performance.
> - $\gamma$ : This hyperparameter is used in the weighted combination of distance $Q_d$ and likelihood $f_n$ to obtain $Q(c’)$, which is used to determine the next pixel location. We introduce $\gamma$ in L210 of the main paper. It maintains a tradeoff between the distance regularizer $Q_d$ and the perturbed likelihood $f_n$. The higher the value of $\gamma$, the greater is the distance regularizer, and consequently the generated path will become closer to that of a straight line. This is not desirable, as a straight line would lose the original composition of the structure. Additionally, because of the perturbation in the likelihood, we do not want the path to go astray. And so, to ensure path completeness, we require $\gamma$ to be non-zero. Through experiments, we obtain the best performance when $\gamma = 0.2$. We provide ablation study results of different $\gamma$ values in Section 12 of the supplementary.
> - $\alpha, \beta$ : These are prior hyperparameters of the Inverse Gamma (IG) distribution which we introduce in L204 of the main paper. We perturb the likelihood using a Gaussian model. As the variance of the Gaussian model is unknown, we use Bayesian probability theory to sample the variance from the IG distribution (its conjugate prior). And so, $\alpha$ is the shape parameter and $\beta$ is the scale parameter of this IG distribution. Ideally we would like a small perturbation of the likelihood and not a strong one. This is because a strong perturbation would corrupt wholly and we would not be able to sample a reasonable skeleton. At the same time, the perturbation should not be too small, otherwise we will not obtain a significant variant. The mean of the IG distribution is $\frac{\beta}{\alpha - 1}$ (when $\alpha > 1, \beta > 0$), which on average is the value of the sampled variance for the Gaussian distribution. We achieve the best performance when $\alpha = 2.0$ and $\beta = 0.01$. The resulting sampled variance for the Gaussian model thus generates reasonable perturbation. We provide ablation study results of different $\alpha, \beta$ values in Section 12 of the supplementary. At the extreme ends of the graph plot in Fig. 13 of the supplementary, the sampled variance is either too low or too high, resulting in a decrease in performance.
>
> We hope the above discussion is helpful. We will definitely add this to the revised version of the paper.
>
> Thank you very much for your review! We hope we were able to clarify your comments, and we would be happy to discuss further!

---

> > ### Comment · Reviewer_ns8Y · 2023-08-13
> >
> > The rebuttal solves all my concerns.

---

> > > ### Author Response · Authors · 2023-08-14
> > > **Thank you for your response!**
> > >
> > > Thank you very much for your response! We are glad that the rebuttal was able to solve all of your concerns. If further clarifications are needed for reevaluating the score, we would be happy to continue the discussion!
> > >
> > > Sincerely,
> > > Authors#14252

---

### Official Review · Reviewer_7bwv · 2023-07-03

**Soundness:** 4 excellent
**Presentation:** 4 excellent
**Contribution:** 4 excellent
**Rating:** 5
**Confidence:** 5

**Summary:**

This paper proposed a topology-aware uncertainty estimation method to segment curvilinear objects. The main contribution focuses on the application of discrete Morse theory (DMT). On several public datasets, the proposed method achieves SOTA performance and the visual results demonstrate the connectivity of vessels or other objects can be enhanced.

**Strengths:**

1. Clear and well-organized paper.
2. The object connectivity is improved by the proposed sound framework.

**Weaknesses:**

1. This paper is mainly based on [24]. The technical contribution is a bit marginal here. Please clearly state the main differences and the take-home insights.
2. Some non-deep methods are also good at achieving better topologies. See:
[1] Liu, Siqi, et al. "Rivulet: 3D neuron morphology tracing with iterative back-tracking." Neuroinformatics 14 (2016): 387-401.
Please include them for discussion and comparison.
3. The title is a bit over-claimed. "Topology-Aware Uncertainty for Curvilinear Object Segmentation" might be more suitable.
4. Using graph models to capture the topology information is not new. See:
[2] Shin, Seung Yeon, et al. "Deep vessel segmentation by learning graphical connectivity." Medical image analysis 58 (2019): 101556.
More discussions and comparison should be added.

**Questions:**

See the above weaknesses.

**Limitations:**

See the above weaknesses.

---

> ### Author Rebuttal · Authors · 2023-08-09
>
> Thank you for your constructive feedback! We will revise our manuscript accordingly. Please find our responses to specific queries below.
>
> **Q1:** This paper is mainly based on [24]. The technical contribution is a bit marginal here. Please clearly state the main differences and the take-home insights.
>
> **A1:** While we tackle the same problem as [24] (i.e., structure-wise uncertainty estimation), our method is significantly different from theirs. We described the key differences in L87 of the main paper, and we elaborate below. We refer to the method in [24] as Hu et al.
> - Hu et al. uses classic DMT to deterministically generate skeletons, thus failing to model intra-structural uncertainty. As we show in Fig. 2a) and Fig. 5 of the main paper, DMT structures often differ from the true GT structure. This is a common problem because of the tortuous nature of the structures. The uncertainty formulation in Hu et. al. mainly relies on the persistence of a structure, thus failing to capture the uncertainty with respect to the structure composition itself. In the ablation study in Table 3, we show how our proposed Prob. DMT results in improvement over DMT. Also as stated in L219, DMT is just one specific instance of our Prob. DMT.
> - We propose a joint inference model to jointly predict uncertainties of all the structures. This joint inference framework avoids explicit enumeration/sampling over the exponential size space of hypotheses. This is in contrast to Hu et al.’s method whose main aim was to generate multiple segmentation hypotheses, thus suffering from the enumeration problem. To reduce this computational burden, they used a global persistence value to prune structures in each run. This pruning was coarse, and such a global thresholding/pruning is harsh in practice, leading to suboptimal uncertainty estimation.
> - Our approach incorporates inter-structural uncertainty using GNNs, recognizing that structures in image space interact with each other and are not isolated. During uncertainty estimation, it is therefore crucial to consider their spatial context, i.e., inter-structural uncertainty. In the Table 3 ablation study, we show how incorporating GNN for inter-structural uncertainty results in improvement.
> - Finally, from Fig. 1, 8, 14 and Tables 1, 6, it is evident that Hu et al.’s method tends to produce over-confident uncertainty estimates --- they assign zero uncertainty (100% confidence) to most structures. On the other hand, our method, accounting for both intra- and inter-structural uncertainties, produces higher fidelity uncertainty estimates.
>
> We believe the above points strongly differentiate our work from [24], both in terms of methodology as well as performance.
>
> **Q2:** Some non-deep methods are also good at achieving better topologies. See: [1] Liu, Siqi, et al. "Rivulet: 3D neuron morphology tracing with iterative back-tracking." Neuroinformatics 14 (2016): 387-401. Please include them for discussion and comparison.
>
> **A2:** Thank you for providing this citation. While Rivulet aims to enhance segmentation quality, our primary goal is structural-level uncertainty estimation, with segmentation improvement coming naturally. Hence directly comparing the two does not seem straight-forward. Furthermore, while Rivulet also generates centerline skeletons similar to DMT, DMT has the important property that it decomposes the likelihood into a set of constituent structures (each structure is a path between a saddle-maxima pair). This decomposition is crucial as we ultimately estimate the uncertainty for each of these structures. Rivulet does not provide any such decomposition and hence cannot be used as a substitute in our framework.
>
> **Q3:** The title is a bit over-claimed. "Topology-Aware Uncertainty for Curvilinear Object Segmentation" might be more suitable.
>
> **A3:** We understand the sentiment and will update the title to "Topology-Aware Uncertainty for Curvilinear Structure Segmentation" in the revised version.
>
> **Q4:** Using graph models to capture the topology information is not new. See: [2] Shin, Seung Yeon, et al. "Deep vessel segmentation by learning graphical connectivity." Medical image analysis 58 (2019): 101556. More discussions and comparison should be added.
>
> **A4:** Thank you for providing this citation. Indeed, the referenced paper uses graph models for vessel segmentation, however, our method utilizes GNNs in a different way. In the referenced paper, the vertices of the graph are pixels sampled from vessel centerlines. In contrast, our method generates a structure-level graph, treating entire structures (collections of pixels) as vertices. Thus we directly model the structures in the graph, instead of a sampled subset of points. Moreover, our graph model generates uncertainty estimates, while the referenced paper focuses on classifying pixels as vessels.
>
> Thank you so much for your review! We would be happy to discuss further!

---

> > ### Comment · Reviewer_7bwv · 2023-08-13
> >
> > Overall, the authors‘ responses solved most of my concerns. Regarding the vessel segmentation tasks, please add the discussion of future works in the final version. For example, the evaluation metric? the main obstacle for clinical applications? Thanks.

---

> > > ### Author Response · Authors · 2023-08-14
> > > **Thank you for your response!**
> > >
> > > Thank you very much for your response! We will definitely incorporate the discussion from the rebuttal to the revised version of the manuscript. We are glad that we were able to solve your concerns. If further clarifications are needed for reevaluating the score, we would be happy to continue the discussion!
> > >
> > > Sincerely,
> > > Authors#14252

---

### Official Review · Reviewer_oww6 · 2023-07-03

**Soundness:** 4 excellent
**Presentation:** 4 excellent
**Contribution:** 4 excellent
**Rating:** 8
**Confidence:** 5

**Summary:**

This paper proposes a novel method for the estimation of uncertainty of the structures in the segmentation results from existing methods/models, in order to facilitate the subsequent proofreading process. To this end, it models the intra-structure and inter-structure uncertainties in two modules. While the former considers the geometry, contrast and model’s confidence, the latter considers neighbours and thus context. The former is implemented using the Probabilistic discrete Morse theory (DMT), which samples the Morse skeletons using the inverse Gamma distribution. The latter uses graph neural network (GNN) for the prediction of the uncertainty of all the structures jointly through regression using the attenuation loss. The proposed method has been validated over three publicly accessible datasets and compared with several state-of-the-art. Relatively better results have been obtained in different metrics. A number of ablation studies have also been carried out on variants of the DMT and GNN and some hyperparameters. Some insights have been accumulated into how the proposed method behaves under different configurations.

**Strengths:**

1.	The topic of the paper is interesting and important for the subsequent validation of the results for image segmentation and find many applications in the read world such as medial image segmentation and analysis, object classification and recognition, industrial quality assurance, etc.
2.	A novel method has been proposed to post-process the segmentation results produced by some existing models for their proofreading and validation. It includes two main modules: intra-structure uncertainty and inter-structure uncertainty estimation. While the former is modelled using the probabilistic DMT, the latter is modelling using the graph neural network. The method is well motivated and supported by solid theory.
3.	The proposed method has been validated over three publicly accessible datasets and compared with several state-of-the-art. Relatively better results have been obtained in different metrics. The t-test has also been performed whether the improvement is significant.
4.	A number of ablation studies have been carried out on the variants of DMT and GNN and some hyperparameters in the process of sampling of Morse skeletons. Some insights have been accumulated into how the proposed method behaves under different conditions, that would instruct how it can be applied in the real world.
5.	The proposed uncertainty estimation has been applied to re-calibrate the segmentation results obtained. The experimental results have shown that it does help identify the false positives and false negatives.


**Weaknesses:**

1.	While the paper targets the proofreading of the segmentation results by experts, its necessity could be emphasised: the automatic algorithms cannot guarantee  the correctness of the segmentation, especially for medical imaging where the anatomy and structures may vary from one subject to another, and their prior knowledge is not always available  This is also the process for the relevant researchers/experts to learn and accumulate insights about the variation of the structures of the subjects for individualised diagnosis, treatment and medicine.
2.	Some details, elaborations, clarifications and discussions are missing from place to place. For example, while the construction of the graph is described later, its main idea could be summarised in Introduction. “structure” is widely used throughout the paper but its count has ever been discussed. While the ground truth is again used to train the GNN, some discussions could be made about its special requirement: GT plays a crucial role in the validation and guiding the estimation of the uncertainty for false positives and false negatives and thus, may have special requirement about quality and reliability. It is not clear how to guarantee this, especially when the datasets are large and include subtle structures. On the other hand, is that possible to use the salient structures to guide the search for faint structures in the spirit of Canny edge detection?  “Persistence value” in L239-240: how to calculate it? where is it from? No details or references are given.
3.	The key steps for the re-calculation of the Morse skeletons in probabilistic DMT may require further investigation. The next point selection criteria should be stated explicitly: maximize the sum of the inverse distance and the likelihood from the candidates and discuss why this criterion is feasible, especially when the distance is not really comparable directly with the likelihood. For the definition of Q(c’), this definition may require further investigation: what is the rationale for this definition? while the first term is inverse distance, is that comparable to the likelihood in the second term? Is this optimal? Is there any other alternative? Is that possible to draw conclusion that the first term will always lie in the unit interval [0, 1]? Overall, this is still a heuristic, which may not hold in some cases. “This process is done separately and in parallel for every structure.” In L222-223: how many Morse skeletons were sampled for each structure and can they be directly used to calculate as uncertainty their means and variances? The final results may be affected by the number of runs and their combinations made.
4.	The computational complexity and time have not been analysed and reported. Thus, it is not clear how efficient the proposed method is and how much time it requires to process a set of given images.
5.	Further analysis of the experimental results would help. For example, while the proposed method is effective in improving the results of the existing methods in ECE, clDice, ARI and VOI, but it is not always in Dice. It is not clear why. Any further insights and explanations would really help.
6.	More ablation studies could be carried out on other parameters and components such as the dimensionality of input feature vector,  crops/bounding boxes on the structure, alternatives to the shortest distance for structure inference.
7.	The claim that the proposed method may be applicable to non-medical applications: civil engineering, road network and railway track segmentation, has not been validated.
8.	It is not clear how the prior knowledge can be used to guide the image segmentation, uncertainty estimation, and result re-calibration, rather than just based on some heuristics in skeleton-recalculation and structure inference, while such heuristics may not hold in some case. Also, it is not clear how such heuristics contribute to the final errors. Some further investigation would be encouraged.


**Questions:**

Please see the comments/concerns in weaknesses,

More detailed comments are as follows:
1.	L41: accept or reject/correct structural proposals efficiently: relative to what: prior knowledge or experience?
2.	L132, each structure: some elaborations would help: what that is: line segment, blob, junction, keypoint, or anything else.
3.	L140-14, At each training iteration, it takes one sample skeleton for each structure: Does this mean that the number of iterations will be determined by the number of the total structures and the number of sampled skeletons for each structure? How to count the number of the structures? any details?
4.	L170, a perturb-and-walk algorithm: refs?
5.	L192-194, At every step, we always walk to the neighboring pixel with the  highest likelihood value: This is a heuristic only, which may not hold in some cases.
6.	L224, The output of Prob. DMT is effectively one sample skeleton: is the same number of sample skeletons as the number of runs? Can all the sample skeletons be directly used to calculate the uncertainties? How to count the number of structures?
7.	L227-228, network that takes as input each structure: This is confusing: takes as input each structure each time or all the structures together? If the former is the case, do you have to run the model many times? what is the motivation for this? Can they be combined together and run the model once to directly estimate the uncertainties?
8.	L237, are smaller crops/bounding boxes: what are the sizes of the boxes? Any details?
9.	Se 3.3: shortest distance: This is heuristic only, which may not hold in some cases.
10.	L299, others -> the others


**Limitations:**

The limitations of the proposed work are discussed in the supplementary materials.

The potential negative social impact are not highly related and thus are not discussed.

---

> ### Author Rebuttal · Authors · 2023-08-09
>
> Thank you for your constructive feedback! We will revise our manuscript accordingly and address your questions below.
>
> **Q1:** Guarantee of GT quality/reliability?
>
> **A1:** We use public datasets which are considered reasonably good, though are not guaranteed to be flawless. Our method can aid in refining the datasets: structure-wise uncertainty can reveal potential annotation errors.
>
> **Q2:** Can salient structures guide search for faint structures like Canny edge detection (CED)?
>
> **A2:** Our work is primarily on uncertainty estimation, tested with human-involved proofreading. An alternative is a human-free automated system that utilizes uncertainty to accept/reject structures. Your analogy of CED suggests high-confidence (low uncertainty) structures could support inclusion of nearby faint (low confidence) ones. This is intriguing and merits further exploration.
>
> **Q3:** Details/ref of Persistence Value L239.
>
> **A3:** We apologize, the missing reference is an oversight on our part. Persistence value (from persistent homology [4]) is defined as the difference of function (likelihood) values of 2 critical cells (saddle-maxima pair). It captures the importance of a structure, thus making it a valuable feature in our framework.
>
> **Q4:** Rationale of $Q(c’)$? Is the first term (inverse distance) comparable to the second term (likelihood)? Is this optimal? Alternatives? Is the first term always in [0,1]? Overall it’s a heuristic which may not always hold.
>
> **A4:** We provide the rationale for $Q(c’)$, especially the inverse-distance term $Q_d$, in L196, L215. As we perform the walk algorithm on a perturbed input, it’s possible the path would go astray and not reach the destination. $Q_d$ acts as a regularizer to likelihood $f_n$, guiding the path from source $c_s$ to destination $c_m$, ensuring path completeness.
>
> $Q_d$ will always be in [0,1] as it’s the inverse of Euclidean distance. Given 2 different pixel locations, the minimum Euclidean distance will be at least 1, and so $Q_d$ will be at most 1. The range of $f_n, Q_d$ are [0,1], so they are comparable. Their combination is weighted by $\gamma$ for the final $Q(c’)$ metric.
>
> Fig.13 (supple.) shows ablation study on $\gamma$ emphasizing $Q_d$’s importance. When $\gamma = 0, Q_d$ is not used, leading to decrease in performance. Notice $\gamma > 0$ results in sharp improvement, empirically showing $Q_d$ is essential in the next point selection criteria $Q(c’)$.
>
> You are correct that $Q(c’)$ is heuristic, but we find it works well in practice. That said, our future goal is to explore a theoretically guaranteed algorithm which can estimate a distribution of Morse complexes from noisy observations.
>
> **Q5: a)** L222: How many skeletons are sampled per structure? Does this equal the no. of runs? Do runs and their combinations influence results? **b)** L140: Does no. of training iterations depend on no. of sampled skeletons? **c)** L227: Can structures be combined to run the model once for direct uncertainty estimation?
>
> **A5: a)** In L294, we mention we take 5 runs, i.e., we sample 5 Morse skeletons per structure. L261 ‘Inference procedure’ outlines how T runs of the framework generate uncertainty estimates. While results vary with run count and resulting combinations, we find that T > 5 did not result in statistically significant improvement. Note that T runs are typical for uncertainty-estimation techniques, including the probabilistic methods we compare against.
>
> **b)** No. of epochs during training is independent of T. Every epoch, we sample 1 skeleton per structure.
>
> **c)** During inference, we conduct T runs, sampling a different skeleton in each run. As you mentioned, the alternative is to pre-generate multiple skeletons at once, and feed them together to obtain uncertainty in one pass. However, we want a flexible framework. If N skeletons are pre-generated, the network will rigidly require N structures, limiting flexibility for users with specific time/memory needs. Our approach lets users tweak T without retraining, unlike the alternative where altering N mandates retraining.
>
> **Q6:** Sec 3.3 shortest distance is a heuristic which may not always hold. Alternatives?
>
> **A6:** In L269, we emphasize that the shortest distance is applied solely to foreground pixels, a stronger constraint than using the shortest distance by itself. This ensures that structures separated by background do not mistakenly assign uncertainties to one another.
>
>  **Q7:** How can prior knowledge guide segmentation, uncertainty and re-calibration, rather than just based on heuristics in skeleton-recalculation and inference?
>
> **A7:**  We are unclear about what you mean by ‘using prior knowledge’. We assume you mean how annotators’ knowledge can be integrated via active learning. Presently, we focus on generating structure-wise uncertainty estimates that can highlight uncertain structures and solicit further input from annotators. How to better learn from user input is a good yet different question that deserves future study. We hope this answers your query, and would be happy to address more comments during the discussion period.
>
> As for heuristics, we believe you are referring to $Q(c’)$. We explain it in Q4/A4, and would like to reiterate that despite a heuristic, we obtain good results in our experimentation.
>
> **Q8:** L41: accept/reject structural proposals: based on prior knowledge or experience?
>
> **A8:** This is subjective: in medical contexts, clinicians rely on expertise, while in non-medical contexts like ROADS [1], an average user can apply their judgment. Thus decisions depend on various factors like context, task, participants, etc.
>
> **Q9:** L132: What is a structure?
>
> **A9:** In L160, we describe structures as Morse structures: V-paths connecting saddle-maxima pairs. A structure is thus a piece of a larger curvilinear structure (Fig.4, 5 visualize some structures that we encounter).
>
> Thank you so much for your review! We would be happy to discuss further!

---

> > ### Comment · Reviewer_oww6 · 2023-08-17
> >
> > Thanks for the detailed responses to the comments raised. While all the datasets for experiments are medical about vessels, and thus some priori knowledge may exist about their distribution, topology, and geometry of eyes and heart. Can such knowledge be used to guide the segmentation or uncertainty estimation process, rather than use the heuristics only to infer the structures?

---

> > > ### Author Response · Authors · 2023-08-17
> > > **Thank you for your response!**
> > >
> > > Thank you very much for your response and the clarification on priori knowledge.
> > >
> > > To the best of our knowledge, there are no well-defined generalizable characteristics of vessels that can be used as priori knowledge. This is because vessel configurations depend on various factors. For instance, the structure of vessels varies significantly across individuals based on age, genetics, health conditions, etc. Pathological conditions can further alter the topology and geometry of vessels. For example, conditions like retinopathy or arteriosclerosis can drastically change the appearance and structure of vessels. Relying on priori knowledge might not always be suitable in such cases.
> > >
> > > Vessels are also not static structures. Their size, branching pattern, and even direction might change based on various physiological factors such as blood flow, oxygen demand, and tissue growth or repair.
> > >
> > > Then there are also technical details: medical datasets can be captured at various resolutions and scales, which influences topology/geometry characteristics.
> > >
> > > Considering the above, it is challenging to obtain generalizable constraints to use as priori knowledge. Solely relying on them can introduce bias or inaccuracies. Thus, we find that while $Q(c')$ is a heuristic, it works well in practice and we obtain good results in our experimentation. Integrating priori knowledge based on the specificities of the dataset would require further investigation.
> > >
> > > Additionally, in this rebuttal, we also conduct an experiment on a non-vessel, non-medical ROADS dataset (as mentioned in 1. under the global response ‘Author Rebuttal by Authors’ above).
> > >
> > > If you have further thoughts on this, we would be glad to continue the discussion!
> > >
> > > Sincerely, Authors#14252

---

> > > > ### Comment · Reviewer_oww6 · 2023-08-18
> > > >
> > > > The clarifications provided are highly appreciated, thanks,

---

### Official Review · Reviewer_qp6i · 2023-07-06

**Soundness:** 3 good
**Presentation:** 3 good
**Contribution:** 2 fair
**Rating:** 5
**Confidence:** 3

**Summary:**


This work aims to contribute to proofreading by proposing uncertain structures in a topological sense. The work proposes a method to quantify a form of structure-wise uncertainty from segmentations, where the framework explicitly models structures as samples from a probability distribution. First, the structures are extracted via discrete Morse theory (DMT). Next, the uncertainty is modeled via a joint prediction model that estimates the uncertainty of a structure in consideration of the surrounding structures. Furthermore, the authors propose a novel probabilistic DMT concept to model intra-structure uncertainty. The method is then successfully experimentally validated.

**Strengths:**

-  The work presents a real generalization of the work on DMT for segmentation [25]. E.g., if I understand it correctly, if one chooses perturbation 0, the result of the method will be exactly the DMT result. This is a nice property and a strong extension of prior work.

- The motivation is clear and interesting.

- The method is well described and formalized.

**Weaknesses:**


1) **Experimentation:** The authors state to propose a "topology-aware" method. However, the authors do not evaluate their results on popular topology-related metrics. E.g., Betti number errors in dimensions 0,1, and 2 (for the 3D dataset). Evaluation of the performance with respect to these metrics will improve the Experimentation. Comparison to the recently published Betti matching error (1), which considers the spatial agreement of the topological structures, would further increase the interpretability of results.


2) **Method** If I understand the method correctly, the DMT calculation at the end could be seen as a post-processing step or additional network to improve connectivity on the results; it has been shown for curvilinear structure segmentation that such an additional step improves segmentation performance. Clearly, such a concept makes this a multi-step procedure which has limitations compared to the cited method by Hu et al.

3) The definition of "topological structures" is not very clear, and to me, it appears to not align with some definitions in algebraic topology, especially in dimension-1. Intuitively I would expect a "topological structure" to be (e.g., in dimension-1) a closed loop. This appears not to be the case here. If I misunderstand this, could the authors clarify how they represent the cycles, and how this is different from their representation of features in dimension-0 and provide more explanations?



**References**:

[] are references from manuscript

(1) Stucki, N. , et al. "Topologically Faithful Image Segmentation via Induced Matching of Persistence Barcodes." ICML (2023).

**Questions:**

This is a minor question regarding the utility of the method. The authors motivate by and mention that their contribution is a way to simplify and improve proofreading. Do you have practical experiments in a proofreading setting that you can share? Is there an experiment with human experts, e.g., readers in ophthalmology?

**Limitations:**

The authors do not provide a dedicated limitations section. I would like to learn more about the limitations of their method in the context of stricter definitions in algebraic topology.

---

> ### Author Rebuttal · Authors · 2023-08-08
>
> Thank you for your constructive feedback! Please find our responses to specific queries below.
>
> **Q1:** The work presents a real generalization of the work on DMT for segmentation [25].
>
> **A1:** We would like to clarify that the goal of this paper is very different from [25]. [25] and other topology-preserving segmentation methods focus only on a segmentation network. In contrast, our method assumes a given segmentation network, and focuses on estimating the uncertainty of the given segmentation output at a structural level. Topology-preserving segmentation is only one of the many applications of our work.
>
> **Q2:** Evaluate results on popular topology-related metrics, e.g., Betti Number errors and Betti Matching error. The comparison would further increase the interpretability of results.
>
> **A2:** Thank you for the suggestion. We now provide these results; please see 2. in the global response ‘Author Rebuttal by Authors’ above.
>
> **Q3:** If I understand the method correctly, the DMT calculation at the end could be seen as a post-processing step or additional network to improve connectivity on the results; it has been shown for curvilinear structure segmentation that such an additional step improves segmentation performance. Clearly, such a concept makes this a multi-step procedure which has limitations compared to the cited method by Hu et al.
>
> **A3:** As we clarify in Q1/A1, given a segmentation network, our goal is to capture the uncertainty of its prediction at a structural level. As the segmentation network is not part of our contribution (and instead is an input to our method), our work cannot be considered as a post-processing/multi-step approach. This is also reflective of real-world scenarios where segmentation networks are often black-boxes, with users being allowed to only access results of the network and not its internals. Thus using only the results of the given segmentation network, our method is able to generate structure-wise uncertainty estimates to streamline the proofreading process.
>
> **Q4:** The definition of "topological structures" is not very clear, and to me, it appears to not align with some definitions in algebraic topology, especially in dimension-1. Intuitively I would expect a "topological structure" to be (e.g., in dimension-1) a closed loop. This appears not to be the case here. If I misunderstand this, could the authors clarify how they represent the cycles, and how this is different from their representation of features in dimension-0 and provide more explanations?
>
> **A4:** This is an excellent question and we would like to clarify the relationship between Morse theory and the theory of persistent homology [4]. In L147, we describe Discrete Morse theory (DMT), and specifically in L159-161, we state that the “structures” are zero- and one-dimensional Morse structures. Morse structures are essentially critical points and special paths connecting them. Morse theory has a very strong relationship with the theory of persistent homology; given a Morse complex, one can exactly compute the persistent homology [5,6]. Discrete Morse theory has been used in the literature for the computation and simplification of persistent homology.
>
> Regarding the representation of cycles/loops (dim-1) in the sense of algebraic topology, while we do not explicitly model them, our method implicitly induces their topological correctness. We express in the paper that our method is topology-aware because of the strong relationship Morse complexes have with topology. When the prediction of the Morse structures is correct, the topology in all dimensions is guaranteed to be correct.
>
> **Q5:** Do you have practical experiments in a proofreading setting that you can share? Is there an experiment with human experts, e.g., readers in ophthalmology?
>
> **A5:** Yes, in L308 of the main paper, we had included results of proofreading experiments comparing our method with Hu et al.’s on the ROSE dataset. Our method was able to improve the segmentation result significantly with a relatively fewer number of clicks. We conducted these experiments with a group of researchers. As future work, we have plans to include clinicians to test our framework.
>
> **Q6:** The authors do not provide a dedicated limitations section. I would like to learn more about the limitations of their method in the context of stricter definitions in algebraic topology.
>
> **A6:** We included the limitations section in Section 14 of the supplementary. As answered in Q4/A4 above, our method uses Morse theory instead of persistent homology, and that there is a strong relationship between the two. As future work, we find that beyond curvilinear segmentation, general object segmentation can also benefit from structure-wise uncertainty (structures in this case would be smaller patches/volumes of the object). Discrete Morse theory can be used in this setting, however, we would need to make use of topological features other than the stable manifold.
>
> Thank you very much for your review! We hope we were able to clarify your comments, and we would be happy to discuss further!

---

> > ### Comment · Reviewer_qp6i · 2023-08-14
> >
> > Dear authors,
> > overall the concerns are appropriately addressed and I recommend accepting the paper. I would really encourage the authors to elaborate more on the theoretical limitations in the manuscript and add the nice additional Experimentation, specifically the performance in terms of Betti matching error, to the main manuscript.

---

> > > ### Author Response · Authors · 2023-08-15
> > > **Thank you for your response!**
> > >
> > > Thank you very much for your response and recommendation! We will definitely incorporate the clarifications from the rebuttal to the revised version of the manuscript. We are glad that we were able to address your concerns.
> > >
> > > Sincerely,
> > > Authors#14252

---

### Author Rebuttal · Authors · 2023-08-09

We thank the reviewers for their time and insightful feedback. We are encouraged that all the reviewers appreciated the novelty of the contribution, and found our work to be methodologically sound and effective.

We have uploaded a 1-page PDF where we add results of additional experiments as requested by the reviewers:

**1. Validate the proposed method on a large non-medical dataset (oww6, ns8Y)**

We conduct additional experiments on ROADS [1] --- a large, non-medical dataset containing 1171 aerial images (1108/14/49 train/val/test), each of 1500 x 1500 resolution. It is a challenging dataset due to obstruction from nearby trees, shadows, varying texture/color of roads, road class imbalance etc.

The quantitative and qualitative results are provided in Table 6 and Fig. 14 of the rebuttal PDF respectively. Table 6 shows that our method outperforms the other probabilistic methods on both ECE and segmentation metrics. Fig. 14 shows that our method generates better fidelity structure-wise uncertainty maps compared to Hu et al. Our heatmaps assign non-zero uncertainty to several false positives/negatives in the backbone UNet’s outputs. This is because we reason about every structure while Hu et al. limits the structure space via pruning.

**2. Evaluate on topology-related metrics: Betti Number error [2] and Betti Matching error [3] (qp6i)**

We provide results on these metrics in Table 7 of the rebuttal PDF. Our method consistently improves the segmentation result in terms of topology. This is consistent with our results in Table 1 of the main paper where our method outperforms the other methods on topology-based metrics like clDice, ARI and VOI. Note that for the 3D PARSE dataset, we were unable to provide Betti Matching error results as its official implementation handles only 2D inputs.

**3. Conduct ablation study on other parameters like dimensionality of input feature vector and crops/bounding boxes (oww6)**

We now include ablation studies on the dimensionality of the input feature vector, and size of the crops/bounding boxes, and report this in Table 8 of the rebuttal PDF. We obtain the best results when the input feature vector size is 32 and the bounding box is 32 x 32. For lower values (16 and 16 x 16), the performance reduces, while for higher values (64 and 64 x 64) we did not observe any statistically significant improvement. Thus to maintain the tradeoff between complexity and performance, we respectively use 32 and 32 x 32 for these hyperparameters.

**4. Please report computational complexity and time (oww6)**

We report the inference time for 5 runs on a 256 x 256 input image patch as follows: Prob.-UNet: 0.196 sec; PHiSeg: 1.811 sec; Hu et al.: 5.485 sec; Ours: 7.433 sec.

The module which takes the most time is the DMT / Prob.DMT computation. Presently, this is the most optimized version as we have implemented it as an external module in C++. We will work towards porting the code to run on GPU to bring down the runtime even more.

Following [7], the computational complexity of DMT is $O(n \log n)$, where $n$ is the number of pixels in the image. Since Prob. DMT additionally computes structure variants, the complexity is $O(n \log n + m)$ where $m$ is approximately the number of foreground pixels, and typically $m << n$ for curvilinear structure datasets. The linear term $m$ is added as we traverse each foreground pixel only once when generating the sample skeleton.

**5. While the proposed method is effective in improving the results on ECE, clDice, ARI and VOI, but not always on Dice. Please provide insights and explanations (oww6)**

This observation is accurate for datasets having curvilinear structures. This is because improvements in segmentation are obtained by recovering broken connections and false negative structures. As curvilinear structures are inherently thin (only a few pixels wide), the recovered connections and false negatives are also quite thin and hence do not affect the Dice score greatly. That being said, we would like to emphasize that we do achieve statistically significant improvement even in Dice for all the 2D datasets (DRIVE, ROSE, ROADS). In the case of the 3D PARSE dataset, we obtain numerically better results for Dice although it is not statistically significant.

**6. L237: What are the sizes of smaller crops/bounding boxes? (oww6)**

We provided these details in L77 of the supplementary. The size of the bounding box was 32 x 32 for 2D datasets, and 32 x 32 x 32 for the 3D dataset. In the Table 8 ablation study, we found this value to have a good tradeoff between computational complexity and performance.

**7. Refs for L170 perturb-and-walk algo? (oww6)**

We give our proposed algorithm the custom name of “perturb-and-walk”, and hence there is no reference. Nonetheless, its design is inspired by the random walk algorithm, for which we cite references in L206.

We further reply individually to each reviewer to address their specific questions.

**References used throughout the rebuttal:**

[1] Volodymyr Mnih. Machine learning for aerial image labeling. University of Toronto (Canada), 2013

[2] Hu, Xiaoling, et al. "Topology-preserving deep image segmentation." NeurIPS, 2019

[3] Stucki, Nico, et al. "Topologically faithful image segmentation via induced matching of persistence barcodes." ICML, 2023

[4] Edelsbrunner, Letscher, and Zomorodian. "Topological persistence and simplification." Discrete & Computational Geometry 28 (2002)

[5] Robins, Vanessa, Peter John Wood, and Adrian P. Sheppard. "Theory and algorithms for constructing discrete Morse complexes from grayscale digital images." TPAMI (2011)

[6] Mischaikow, Konstantin, and Vidit Nanda. "Morse theory for filtrations and efficient computation of persistent homology." Discrete & Computational Geometry 50 (2013): 330-353

[7] Dey, Tamal K., Jiayuan Wang, and Yusu Wang. "Graph reconstruction by discrete Morse theory." arXiv preprint arXiv:1803.05093 (2018)

---

### Decision · Program_Chairs · 2023-09-21

**Decision:**

Accept (poster)

**Comment:**

This paper proposes a method for estimating segmentation uncertainty for segmentation of topological structures in images, to be used to "proof-read" the segmentations in a postprocessing step. This is important, as topological structures can easily be broken by a few incorrectly segmented pixels. The paper was appreciated by the reviewers already before rebuttal, and the authors' rebuttal further responded to reviewer concerns.

In their final version, the authors should consider the reviewer comments carefully to improve the quality of their paper. The authors should also follow up on all promised improvements. The authors should also make sure to discuss the limitations of their work in the main paper, not just the supplements.